# *In vitro* mechanical vibration down-regulates pro-inflammatory and pro-fibrotic signaling in human vocal fold fibroblasts

**David Hortobagyi** [iD]*, **Tanja Grossmann, Magdalena Tschernitz, Magdalena Grill** [iD],
**Andrijana Kirsch, Claus Gerstenberger, Markus Gugatschka**

Division of Phoniatrics, Medical University of Graz, Graz, Austria

* david.hortobagyi@medunigraz.at

## Abstract

### Introduction

Voice rest following phonotrauma or phonosurgery has a considerable clinical impact, but clinical recommendations are inconsistent due to inconclusive data. As biopsies of the vocal folds (VF) for molecular biology studies in humans are unethical, we established a new *in vitro* model to explore the effects of vibration on human vocal fold fibroblasts (hVFF) in an inflammatory and normal state, which is based on previously published models.

### Methods

By using a phonomimetic bioreactor we were able to apply predefined vibrational stress patterns on hVFF cultured under inflammatory or normal conditions. Inflammatory and pro-fibrotic stimuli were induced by interleukin (IL)1β and transforming growth factor (TGF)β1, respectively. Mechanical stimulation was applied four hours daily, over a period of 72 hours. Outcome measurements comprised assessment of extracellular matrix (ECM)-related components, angiogenic factors, and inflammatory and fibrogenic markers on gene expression and protein levels.

### Results

Under inflammatory conditions, the inflammatory cytokine IL11, as well as the myofibroblast marker alpha smooth muscle actin (α-SMA) were significantly reduced when additional vibration was applied. The desirable anti-fibrotic ECM component hyaluronic acid was increased following cytokine treatment, but was not diminished following vibration.

### Conclusion

Our experiments revealed the effect of vibrational stress on hVFF in an inflammatory state. Elevated levels of certain pro-inflammatory/pro-fibrotic factors could be mitigated by additional vibrational excitation in an *in vitro* setting. These findings corroborate clinical studies which recommend early voice activation following an acute event.

**Data Availability Statement:** All relevant data are within the paper and its Supporting information files.

**Funding:** The authors received no specific funding for this work.

**Competing interests:** The authors have declared that no competing interests exist.

# Introduction

## Inflammation

Disruption of the vocal fold (VF) mucosa caused surgically, or mechanically by phonotrauma lead to an upregulation of inflammatory cytokines and enzymes as well as damage associated proteins (DAMP) in the VF [1–3]. Dermal studies have shown that interleukin (IL) 1 is the body's first signal of tissue damage. This leads to a degranulation of thrombocytes and consequently to a release of different growth factors such as transforming growth factor β (TGF β), platelet-derived growth factor (PDGF) and epidermal growth factor (EGF). These factors in turn activate a plethora of other cell types, thereby inducing inflammation and wound healing [4]. Since these processes increase the energy metabolism of the tissue, a sufficient blood and consequently nutrient supply is essential. This is enabled by angiogenic factors, which contribute to vessel formation [5].

An *in vivo* study in rats by Lim et al. suggests that these processes are similar in VF [6]. These inflammatory processes have a significant impact on VF metabolism and tissue remodeling and may result in VF scarring, with permanent impairment of the viscoelastic properties and dysphonia. Understanding and modelling these processes could be a useful input into clinical practice guidelines regarding recommendations of voice rest following phonosurgery or phonotrauma.

## Vocal rest

Voice therapy can prevent the development of benign VF lesions and, in some cases (e.g. VF nodules), can also function as primary treatment. However, in the majority of lesions, it acts primarily as an adjunctive therapy to a surgical intervention [7, 8]. In these cases, it is still not clear when it is best to begin with voice therapy, though Tang et al. showed that even a presurgical application has positive effects on subjective voice parameters [9].

Endolaryngeal surgery of the VF is an established treatment option for benign and malignant VF lesions and is nowadays routinely performed in many countries of the world. While many lesions can be successfully treated surgically, there is still uncertainty about the postoperative management, and especially the role of voice rest. The recommendation on postoperative voice rest is based on the hypothesis that preserving the tissue improves wound healing. A distinction is made between absolute and relative voice rest. The former entails a complete avoidance of any mechanical stimulation of the VF tissue through phonation. The latter still lacks a clear definition. But as in other medical fields, where a rapid mobilization after surgical interventions is now often recommended to achieve an early recovery, there has been a tendency over the past years, to decrease the interval of voice rest following phonomicrosurgery [10, 11]. Obviously, the objectives of rehabilitation of, for example, a joint and the VF are not identical. While resilience is more important in orthopedics, pliability is the predominant aim in VF wound healing. Nevertheless, early mobilization is known to have antifibrotic effects, which might also be useful in the field of laryngology [12]. Furthermore, a shorter period of voice rest means that voice therapy can be started sooner.

Recent randomized controlled studies could not confirm any beneficial effects of absolute voice rest relative to a reduced postoperative voice use, as measured in terms of subjective parameters (e.g. VHI) as well as acoustic and aerodynamic parameters [13, 14]. There are still different concepts dealing with this issue, and a commonly accepted consensus is not available [15–21]. One reason for this is that it is virtually impossible to follow postinterventional changes on the cellular level in humans, as the VF are anatomically a small structure and every biopsy carries the risk of scarring. However, in order to reach a deeper understanding of VF

physiology and pathophysiology it is of the utmost importance to explore mechanisms on a molecular level.

## Bioreactors

For this purpose, various bioreactors, trying to imitate phonatory stresses *in vitro*, were developed in the past years. Farran et al. and Kim et al. built bioreactors where cells were seeded onto flexible membranes which were then set into vibration either by an air stream or by linear actuators, respectively. Since the cells in these models are single-layered, these devices are described as 'two-dimensional' reactor types [22, 23].

Besides these 'two-dimensional' models, there are some three-dimensional approaches as well. Gaston et al. and Titze et al., for example, developed three-dimensional scaffolds where cells are predominantly exposed to tensile stresses [24, 25].

Conversely, Latifi et al. developed a bioreactor, which is currently closest to the *in vivo* situation. Their intention was to imitate all mechanical forces, VFF are exposed to during phonation. They injected a scaffold, consisting of human VF fibroblasts (hVFF), into silicone VF replicas. Cells were exposed to mechanical forces by applying an airflow from beneath which creates a similar mode of vibration to that of the human VF in the larynx [26].

A special phenomenon of fibroblasts in general is their diversity. They can accomplish a plethora of functions depending on their localization [27]. The studies mentioned above support the idea that fibroblasts respond to vibration by altering their extracellular matrix (ECM) production. The VFF is predominantly responsible for the composition of the lamina propria. Additionally, in a recent study, VFF were shown to be one of the major sources of high-mobility group box 1, an important DAMP [3]. This cell type is, therefore, essential for a smooth phonation as well as for wound healing.

Zhang et al. demonstrated that the application of cyclic tensile strain on hVFF attenuates the inflammatory reaction induced by cigarette smoke [28]. However, to our knowledge, the impact of vibration on hVFF during inflammation is still unclear. The lack of suitable devices to mimic these conditions has hindered such efforts so far.

Our recently designed phonomimetic bioreactor allowed us to apply specific vibrational patterns to hVFF in culture [29]. Using this device, we were able to study cellular responses of hVFF following a pro-fibrotic and inflammatory stimulus under static and dynamic conditions for the first time. By combining these factors, we hope to mimic inflammation after phonosurgery as accurately as possible, in order to better understand the effects of vibration versus (voice) rest on ECM-related molecules and angiogenic factors as well as on inflammatory and fibrogenic markers.

## Materials and methods

### Cell culture and treatment

Immortalized hVFF, kindly provided by Prof. S. Thibeault (University of Wisconsin-Madison, USA), were cultivated in standard medium (SM) consisting of Dulbecco's modified Eagle's medium (DMEM, 4.5g/L glucose, Sigma Aldrich, Vienna, AT) supplemented with 10% FBS (Biowest, Nuaillé, FR) and 0.2% Normocin (Invivogen, San Diego, CA, USA) as previously described [30]. The mean number of cell passages of the four experiments was 125. Cells were seeded at a density of 144000 cells per well on two flexible-bottomed BioFlex culture plates (Flexcell International, Burlington, NC, USA), unless indicated otherwise. The Bioflex silicone membranes were coated with Pronectin, a fibronectin-like engineered protein. The coating was selected during previous projects, as cell morphology was identical to standard culture conditions. 24 hours later, the medium was changed to a serum-free medium for starvation.

After another 24 hours, cells were divided into four groups exposed to different conditions (static or dynamic–with or without cytokines). The starvation medium was replaced with basal medium (BM) consisting of DMEM, 0.5% FBS, 0.2% Normocin, and 100μM ascorbic acid (all Sigma Aldrich). Pro-fibrotic and inflammatory culture conditions were achieved by cytokine treatment (addition of TGF-β1 and IL1β, (both at 5ng/mL, R&D Systems, Minneapolis, MN, USA) to the BM. This combination and these concentrations of cytokines were used based on previous publications [31–34] and preliminary trials by our group, in which cytokines were capable of causing durable changes in the expression of other cytokines and ECM-related proteins for at least 72 hours (see S1 File and S1 Fig). Furthermore, adding inert macromolecules to the medium enhances the deposition of ECM components, such as collagen. This is a crucial point in studying fibrogenesis [35, 36]. Therefore, all experiments were performed under crowded conditions. For this purpose, BM was supplemented with macromolecules, namely a mixture of 37.5mg/mL 70 kDa Ficoll (Fc) (Sigma-Aldrich St. Louis,MO,USA) with 25mg/mL 400kDa Fc5, 400μL L-Glutamine 1x and 100μM ascorbic acid [35].

Cells assigned to the static group were kept in a separate incubator. Vibration was applied for four hours daily as described below. The cells were harvested after 73 hours (72 hours of treatment/control conditions plus one hour of rest (Fig 1)). Subsequently, harvested material was processed for qPCR, Western Blot, Luminex and ELISA as described below in detail. The main focus was on ECM-related proteins, proangiogenic factors and proinflammatory as well as profibrotic markers. The selection of the different genes and molecules is justified below.

All experiments were repeated four times.

## Selection of genes and molecules

**ECM-related molecules.**   ECM-related molecules describe components of the extracellular scaffold and enzymes that are responsible for the ECM homeostasis and therefore essential in scar development. Two functionally very important ECM components of the VF lamina propria are collagen and HA.

Procollagen, the progenitor molecule, is secreted from the cell and undergoes modifications that lead to the formation of a crosslinked network of fibrils. The most abundant types of collagen in the VF lamina propria are type 1 and type 3. Matrix metalloproteinase (MMP) 1 is responsible for the degradation of collagen type 1 and 3. Tissue inhibitor of metalloproteinase (TIMP) on the contrary is considered to be the antagonist of MMP [37, 38].

Hyaluronic acid is a ubiquitous glycosaminoglycan and is of particular importance for tissue. It is synthesized via hyaluronan synthases (HAS) of which three isoenzymes HAS1, HAS2 and HAS3 with different kinetic properties are known. Hyaluronidase (HYAL) 2 is responsible for HA degradation [39, 40].

Additionally, fibronectin (FN) was included in the present investigations because it was known to play an important role in cell-to-cell and matrix-to-cell adhesion [41].

**Proangiogenic factors.**   A previous investigation of the effect of cigarette smoke extract on VFF by our group showed a significant upregulation of vascular endothelial growth factors (VEGF) A and C [42]. In order to evaluate potential changes on VEGF A and C following cytokine treatment and vibration, these two factors were included in the present study.

**Inflammatory and fibrogenic markers.**   In this study it was only possible to investigate a limited subset of cytokines, in the knowledge that many more could be relevant.

Alpha smooth muscle actin *(ACTA) 2* is the encoding gene for the myofibroblast marker α-SMA. This protein contributes to VF tissue contraction and thus to VF fibrogenesis [43].

IL1β, IL6, cyclooxygenase (COX) 2 and TGFβ1 were selected since various studies have previously demonstrated their decisive role in acute inflammatory reactions [6, 44].

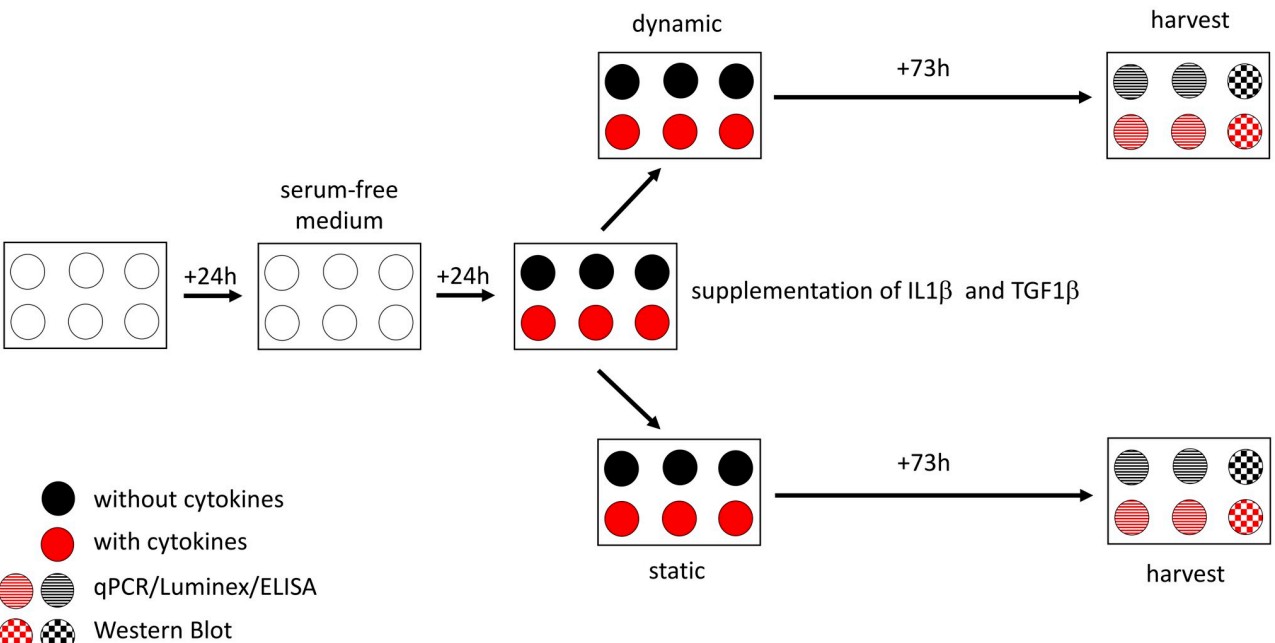

**Fig 1. Experimental design.** Twenty-four hours after seeding the cells, the cell culture medium was changed to a serum-free medium for starvation. The next day, cells were divided into four groups with different conditions (with or without cytokines–static or dynamic). After a further 73 hours, cells were harvested. Two wells of each condition were used for qPCR, Luminex as well as ELISA and one for Western Blot. All samples were taken into account for LDH assay.

In *in vitro* experiments, basic fibroblast growth factor (bFGF) had an anti-inflammatory effect by downregulating collagen production and upregulating HA production [45].

IL11, another member of the IL6 family, is known to be mechanoresponsive and is also involved in tissue fibrogenesis. It has been shown that this cytokine is a downstream effector of TGFβ1 and contributes to the differentiation of pulmonary fibroblasts to *ACTA2* positive myofibroblasts [46–48].

## Mechanical stimulation

Mechanical stimulation was performed using a phonomimetic bioreactor developed and established by our group. In short, the flexible bottomed 6-well plates were oscillated by aerodynamic pressure fluctuation induced by a loudspeaker beneath the plate [29, 49].

Immortalized hVFF were exposed to vibration for three days, as this time period corresponds to the acute inflammatory phase [1, 18]. The stimulatory pattern, a linear sinusoidal chirp sound in a frequency range of 50 Hz to 250 Hz corresponds to the frequency range of the primary larynx sound in humans. The membrane displacement in the center of a well was dependent on the applied frequency and averaged 82 μm ± 5 μm. The input voltage (V) on the loudspeaker was set to 1.1 V and served as a measure of the vibration intensity. Stimulation was applied for four hours per day, following the publication of Titze et al. [50] There are a handful of publications dealing with displacements (vertical and horizontal) of human VF during phonation, reporting a wide range of values (depending on the setting and/or measurement method). The simulation of these displacements is reproducible with our bioreactor and complies with the maximum displacement in humans [51]. In order to rule out any interferences, cells assigned to the static group were kept in a separate incubator.

## Lactate dehydrogenase (LDH) assay

Cell viability was assessed by measuring lactate dehydrogenase (LDH) activity in the supernatants, using the Pierce™ LDH Cytotoxicity Assay Kit (Thermo Scientific). To evaluate maximum LDH activity, cells were seeded in parallel in two wells of a 24-well plate at the same density parallel to the 6-well-plates. Medium was changed in the same way as for the experimental cells. Additionally, SM ± cytokines without any cells was added to another two wells serving as blanks. 30 minutes before harvesting, 10x lysis buffer was added to the cell-seeded wells followed by further incubation at 37°C. Subsequently, 50μL supernatants of all wells were collected and processed in duplicates according to the manufacturer's instructions. Absorbances at 490nm and 680nm were measured using the Spectramax Plus 384 Microplate Reader (Molecular Devices, San Jose, CA, USA). The differences between these absorbances were computed and the mean values of the duplicates calculated. The blank values were then substracted from each sample. The LDH activity was expressed as percentage of the maximal LDH activity.

## RNA isolation and Reverse Transcription-quantitative Polymerase Chain Reaction (RT-qPCR)

Four out of six wells per plate were employed for RT-qPCR. Cells were lysed with QIAZOL Lysis Reagent (Qiagen, Hilden, GER) followed by RNA extraction with the miRNeasy Mini Kit (Qiagen, Hilden, GER) according to the manufacturer's instructions. RNA concentrations were determined on a NanoDrop 2000c UV–Vis spectrophotometer (Thermo Scientific, Waltham, MA). RNA purity was established by calculating the ratio of the absorbance at 260nm and 280nm. $A_{260}/A_{280} > 1,8$ was considered to be clean. 1μg total RNA of each sample was reverse transcribed to cDNA using the QuantiTect Reverse Transcription Kit (Qiagen) according to the manufacturer´s instructions. 10ng in 4μL of cDNA were mixed with 6μL of Primer-SYBR Master Mix (Promega/USA) consisting of 1μL (200nM each) Primer F&R and 5μL GoTaq Master Mix (Promega/USA). RT-qPCR was performed in technical triplicates using the LightCycler 480 system (Roche, Vienna, AT). The following program was applied for denaturation, amplification and the melting curve analysis, respectively: 2 min/95°C, 45 cycles of 10 s/95°C and 1 min/60°C, followed by constantly increasing the temperature at a rate of 2.5°C/min from 55°C to 95°C [52]. $C_T$ values were calculated with the AbsQuant/$2^{nd}$ Derivative Max method of the LightCycler 480 software and relative quantification of expression levels was performed using the geometric mean of beta-2 microglobulin (*B2M*) and ubiquitously expressed transcript protein (*UXT*) as internal references [53, 54]. Validation of the housekeeping genes was done as previously described by Dheda et al. [55] Subsequently gene expression of ECM-related proteins *HAS1*, *HAS2*, *HAS3*, *HYAL2*, *COL1A1*, *COL1A2*, *COL3A1*, *FN1*, *MMP1* and *TIMP1* and pro-inflammatory/pro-fibrotic markers (*ACTA2*, *IL1β*, *IL6*, *IL11*, *COX2* and *TGFβ1*) was evaluated. Primer sequences are listed in Table 1.

## Western blot analysis

Intracellular changes of collagen and α-SMA, a myofibroblast marker, were measured by Western Blot [43]. From each plate two wells (with or without cytokines) were used for the collection of protein lysates. Cells were washed with ice-cold PBS and lysed with ice-cold RIPA buffer (Cell Biolabs San Diego, CA, USA) supplemented with 1x HaltProtease and Phosphatase Inhibitor Cocktail and 5mM EDTA solution (both from Thermo Fisher Scientific). Total protein concentration was determined using the Pierce BCA Protein Assay Kit (Thermo Fisher Scientific) according to the manufacturer's instructions. 25μg of total protein were mixed with

**Table 1. Primer sequences used for RT-qPCR.**

| Gene | Gene symbol | Forward primer | Reverse primer |
| --- | --- | --- | --- |
| Alpha smooth muscle actin | ACTA2 | CGTTACTACTGCTGAGCGTGA | GCCCATCAGGCAACTCGTAA |
| Beta-2-microglobulin | B2M | AGGCTATCCAGCGTACTCCA | CGGATGGATGAAACCCAGACA |
| Collagen I α1 | COL1A1 | CCCCGAGGCTCTGAAGGT | GCAATACCAGGAGCACCATTG |
| Collagen I α2 | COL1A2 | ACCACAGGGTGTTCAAGGTG | CAGGACCAGGGAGACCAAAC |
| Collagen III α1 | COL3A1 | GACCTGGAGAGCGAGGATTG | GTCCATCGAAGCCTCTGTGT |
| Prostaglandin-endoperoxide synthase 2 | PTGS2/COX2 | AGTGCGATTGTACCCGGACAGGA | TGCACTGTGTTTGGAGTGGGTTTCA |
| Fibronectin 1 | FN1 | CTGCAAGCCCATAGCTGAGA | GAAGTGCAAGTGATGCGTCC |
| Hyaluronan synthase 1 | HAS1 | CTTCCTAAGCAGCCTGCGAT | TATATAGGCCTAGAGGACCGCTG |
| Hyaluronan synthase 2 | HAS2 | ATGCTTGACCCAGCCTCATC | TTAAAATCTGGACATCTCCCCCAA |
| Hyaluronan synthase 3 | HAS3 | ATCATGCAGAAGTGGGGAGG | GAGTCGCACACCTGGATGTA |
| Hyaluronidase 2 | HYAL2 | CGTGGTCAATGTGTCCTGGG | CCCAGGACACATTGACCACG |
| Interleukin 1 beta | IL1β | GATGGCTTATTACAGTGGCAATGA | GGTCGGAGATTCGTAGCTGG |
| Interleukin 6 | IL6 | AACCCCCAATAAATATAGGACTGGA | CCGAAGGCGCTTGTGGA |
| Matrix metalloproteinase 1 | MMP1 | CACGCCAGATTTGCCAAGAG | GTTGTCCCGATGATCTCCCC |
| Metallopeptidase inhibitor 1 | TIMP1 | GGAATGCACAGTGTTTCCCTG | GGAAGCCCTTTTCAGAGCCT |
| Transforming growth factor beta 1 | TGFβ1 | TACCTGAACCCGTGTTGCTC | GCTGAGGTATCGCCAGGAAT |
| Ubiquitously expressed transcript protein | UXT | GCAGCGGGACTTGCGA | TAGCTTCCTGGAGTCGCTCA |

the appropriate amount of 4x Laemmli Buffer (Bio-Rad, Hercules, CA, USA) and DTT and boiled for 5 min at 95°C. These samples were then subjected to SDS-PAGE using 4–20% Mini PROTEAN TGX gels (Bio-Rad), after which the proteins were blotted onto a nitrocellulose membrane (Bio-Rad). The blots were blocked in 5% milk for two hours, followed by incubation with the primary antibody against α-SMA (Sigma-Aldrich, A5228, 1:1000); COL1A1 (Nordic Bio Site, Täby,SE, ABB-2670, 1:2000) or glyceraldehyde phosphate dehydrogenase (GAPDH, Cell Signaling, Danvers, MA, USA, #2118S, 1:5000) over night at 4°C. After washing, the blots were incubated with the appropriate secondary antibody Goat Anti-mouse IgG H&L (Abcam, Cambridge, UK, abb6789, monoclonal, 1:5000 for α-SMA) and Goat Anti-rabbit IgG H&L (Abcam, abb6721, polyclonal, 1:5000 for COL1A1 and GAPDH). Signal was detected using the SuperSignal West Pico Chemiluminescent Substrate (Thermo Fisher Scientific) and images were acquired with the ChemiDoc Touch system (Bio-Rad). Densitometric analyses were conducted using ImageLab software (Bio-Rad).

## Magnetic Luminex assay

Human Magnetic Luminex® Assays (LXSAHM, R&D Systems) were used for the quantification of selected proteins in the supernatant. The custom-designed multiplex assays measuring COL1A1, FN1, TIMP1, proangiogenic factors (VEGF A, VEGF C), bFGF and IL11 were performed according to the manufacturer's protocol. Standards for each analyte were provided by the manufacturer and included in the kit. These standards were used to prepare standard curves (see S4 File). Additionally, SM without cytokines and SM with cytokines was used as blank, respectively. The sample dilutions for the selected markers were estimated from previous Luminex measurements (e.g. Gugatschka et al 2019, Grossmann et al. 2020). For COL1A1, FN1 and TIMP1 measurement, samples were pre-diluted 1:50; to determine all other analytes, samples were undiluted. The assays were measured on the Bio-Plex 200 assay reader (Bio-Rad). Calculations were carried out using the Bio-Plex Manager Software Version 6.2 (Bio-Rad).

## Enzyme-linked immunosorbent assay (ELISA)

Hyaluronic acid (HA) levels in the supernatant were determined using the sandwich enzyme immunoassay Quantikine® ELISA Kit (DHYAL0, R&D Systems) according to the manufacturer's protocol. Briefly, an HA standard provided in the kit was used and diluted with calibrator diluent RT5-18. This diluent also served as the zero standard. An HA standard curve was prepared with HA concentrations from 0.625 to 40 ng/mL. SM without cytokines and SM with cytokines were used as blanks. All samples and blanks were pre-diluted 1:100 with Calibrator Diluent RD5-18 and assayed with HA standards on the same plate in technical duplicates. Optical absorbance (OA) values were measured using the Spectramax Plus 384 Microplate Reader (Molecular Devices), OA values at 540nm were substracted from OA values at 450nm. Sample concentrations were determined from calibration curve, the corresponding blank value was subtracted and resulting numbers were multiplied with the dilution factor (x100).

## Statistical analysis

Statistical analysis was performed with SPSS (Version 25). To determine normal distribution, the Shapiro-Wilk test was performed. The four different conditions, static without cytokines, static with cytokines, vibration without cytokines and vibration with cytokines were treated as independent variables. After the proof of normal distribution, one-way ANOVA and, in cases of significance, a Tukey post-hoc test was used to evaluate significant differences between the different conditions. If the values were not normally distributed, a Kruskal-Wallis test was performed, followed by a pairwise comparison using a Mann-Whitney-U test. A p-value $< 0.05$ was determined as statistically significant and values are represented as means ± standard deviation (S.D.). Partial eta squared ($\eta^2$) was calculated to determine the effect size.

# Results

## Fibroblast viability

No significant differences in LDH-activity were found between the different conditions ($p = 0.966$). (Fig 2).

## Gene expression and protein synthesis

**ECM-related molecules and angiogenic factors.** Statistical tests revealed significant differences in HA ($p = 0.008$; $\eta^2 = 0.850$), *HAS1* ($p = 0.043$; $\eta^2 = 0.243$) and *HAS3* ($p = 0.000$; $\eta^2 = 0.792$). Post hoc analysis of HA concentrations in supernatants showed a significant upregulation with cytokine treatment, while vibration did not show an additional effect (Fig 3a). Gene expression results were consistent with significantly elevated transcript levels of *HAS1* and *HAS3* due to cytokine treatment compared to those wells without cytokine exposure. (Fig 3b and 3d). Gene expression of *HAS2* showed a similar trend without any significance ($p = 0.125$; $\eta^2 = 0.369$) (Fig 3c). *HYAL2* remained unchanged ($p = 0.155$; $\eta^2 = 0.343$) (Fig 3e).

One-way ANOVA showed significant alterations of *COL1A1* ($p = 0.000$; $\eta^2 = 0.825$); *COL1A2* ($p = 0.000$; $\eta^2 = 0.795$) and *COL3A1* ($p = 0.005$; $\eta^2 = 0.640$). Tukey post hoc comparison revealed a statistically significant upregulation of all the three *COL* subtypes with cytokine treatment but without any changes due to vibration (Fig 4a–4c). Intracellular protein content of COL1A1 showed a similar trend but did not meet statistical significance ($p = 0.076$; $\eta^2 = 451$) (Fig 4d) whereas for protein content in the supernatant a significant difference was found. In pairwise comparisons, the cytokine treatment significantly increased COL1 concentration compared to the static non-inflammatory group (Fig 4e). Matrix metalloproteinase1 (*MMP1*) expression was significantly altered ($p = 0.009$; $\eta^2 = 0.604$). In the post hoc test,

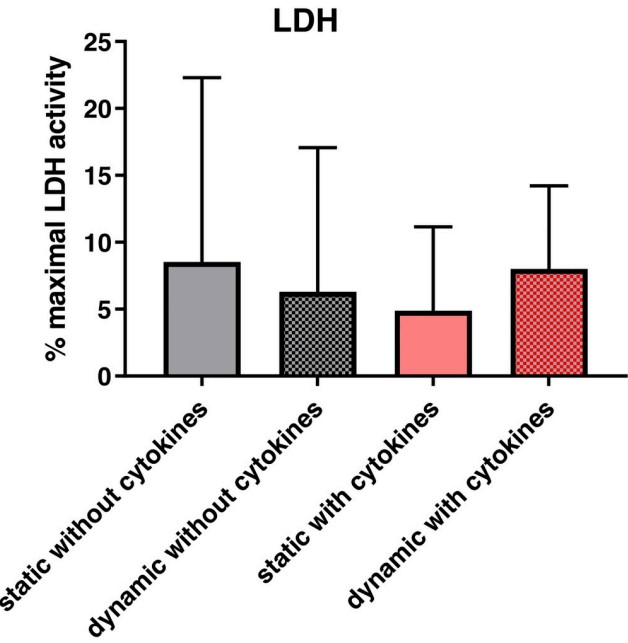

**Fig 2. Effect of vibration and/or cytokine treatment on cell viability and gene expression.** After 72 hours of exposure to a vibration pattern and/or cytokine treatment and an additional one-hour rest period, supernatants were collected for LDH activity assay. The LDH activity was expressed as percentage of the maximal LDH activity. p<0.05 was considered significant.

downregulation was observable after the application of cytokines compared to the groups without cytokine exposure (Fig 4f). Neither condition influenced *TIMP1* expression (p = 0.629; $\eta^2$ = 0.130) (Fig 5c).

Statistical tests allowed the null hypothesis to be rejected concerning *FN1* gene expression (p = 0.000; $\eta^2$ = 0.804) as well as FN1 protein concentration (p = 0.012; $\eta^2$ = 0.790). In post hoc analysis, gene expression was significantly increased by cytokine treatment and was not altered by additional vibration (Fig 5a). Protein levels were similarly upregulated by cytokine treatment under static condition, but did not undergo any significant further change on combination with vibration (Fig 5b).

Protein content of bFGF in supernatant was significantly altered (p = 0.000; $\eta^2$ = 0.859). Tukey post hoc analysis showed a significant increase upon cytokine treatment. (Fig 5d). Proangiogenic factors VEGF A (p = 0.763; $\eta^2$ = 0.089) and VEGF C (p = 0.481; $\eta^2$ = 0.180) were not altered by any treatment on protein level (Fig 5e and 5f).

**Inflammatory and fibrogenic markers.** One-way ANOVA revealed significant changes of *IL1β* (p = 0.016; $\eta^2$ = 0.564), *IL6* (p = 0.003; $\eta^2$ = 0.670), *TGF-β1* (p = 0.012; $\eta^2$ = 0.587), *ACTA2* (p = 0.001; $\eta^2$ = 0.754) gene expression and IL-11 (p = 0.000; $\eta^2$=) protein concentration as well as α-SMA (p = 0.000; $\eta^2$ = 0.798) protein levels. Pairwise comparisons of the gene expression of *IL1β* only showed a significant upregulation with the combination of cytokine treatment and vibration compared to the non-inflammatory groups (Fig 6a). While *IL6* was significantly upregulated after cytokine exposure, no further change was detected with vibration (Fig 6b). Protein analysis of IL11 in the supernatant showed a significant increase following cytokine treatment, which was significantly reduced by vibration (Fig 6c).

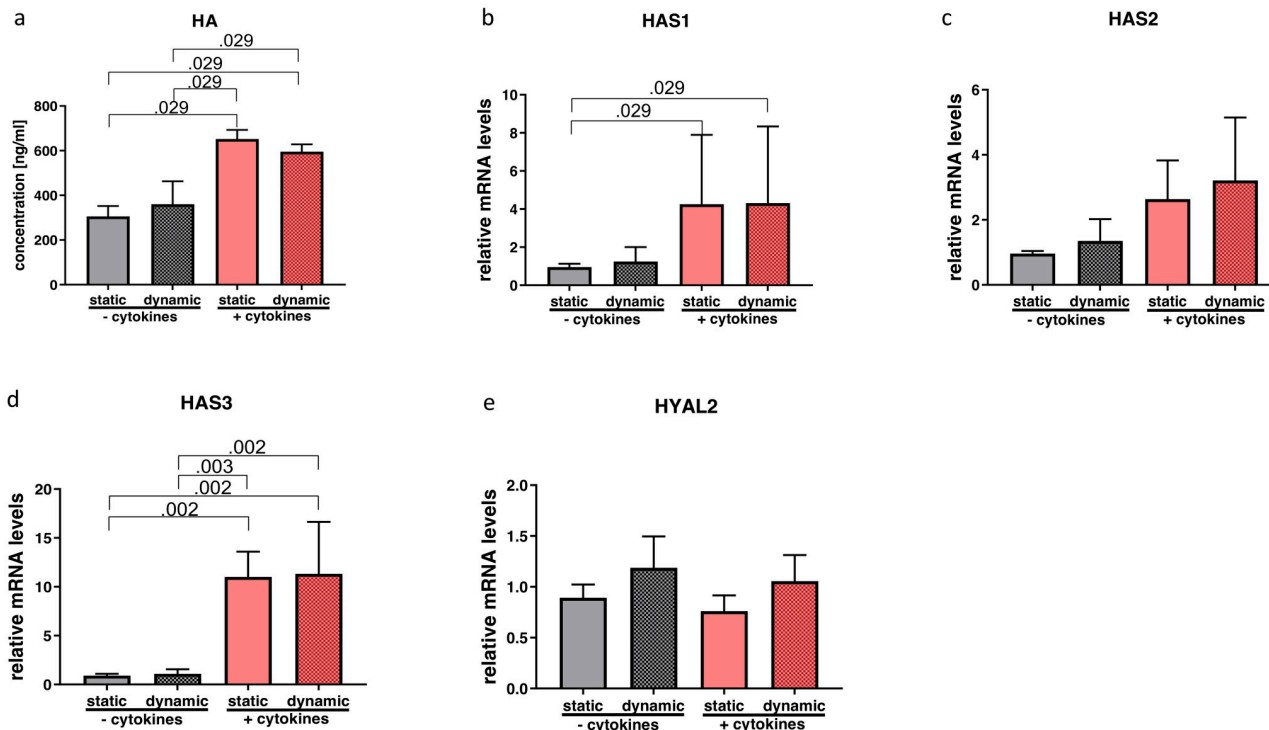

**Fig 3. Effect of cytokine treatment and/or vibration on HA metabolism.** Supernatants were collected for ELISA to quantify values of HA (a). RT-qPCR analysis of HA-related gene expression (b-e) was performed. Results are represented as mean ± S.D. of four independent experiments (N = 4). Statistical analysis was performed using one-way ANOVA (normally distributed data) or Kruskal-Wallis test (non-parametric data), p<0.05 was considered significant; decimal numbers above the bars represent the statistically significant p-values from post-hoc comparisons. For the p-values not presented here, see S2 File. HA (hyaluronic acid), *HAS1* (hyaluronan synthase 1), *HAS2* (hyaluronan synthase 2), *HAS3* (hyaluronan synthase 3), *HYAL2* (hyaluronidase 2).

The gene expression profile of *TGF-β1* was significantly increased by cytokine treatment in the static condition. The combination of cytokine exposure and vibration however, did not yield a significant change (Fig 6d). Likewise, cytokine treatment alone showed a statistically significant upregulation of *ACTA2* gene expression that was not altered by vibration (Fig 6e). In contrast, significantly upregulated protein content of α-SMA through cytokine treatment was reduced by vibration (Fig 6f).

*COX2* gene expression did not show significant alterations in any condition (p = 0.506; η² = 0.027).

## Discussion

The early phase of tissue repair is accompanied by an inflammatory reaction, which among other things promotes wound healing, recruits inflammatory cells and can preempt an infection in the region of injury [1]. However, this physiological process bears the risk that inflammatory remodeling of VF tissue may impair the viscoelastic properties of the lamina propria and in the worst case could lead to persistent dysphonia due to VF scarring [56]. For this reason it is essential to manage the post-injury wound healing phase as well as possible. Understanding the biological impact of vibration on hVFF in an inflammatory setting is the first step towards this goal, and relates directly to the clinically highly relevant questions of voice rest following phonotrauma or phonosurgery.

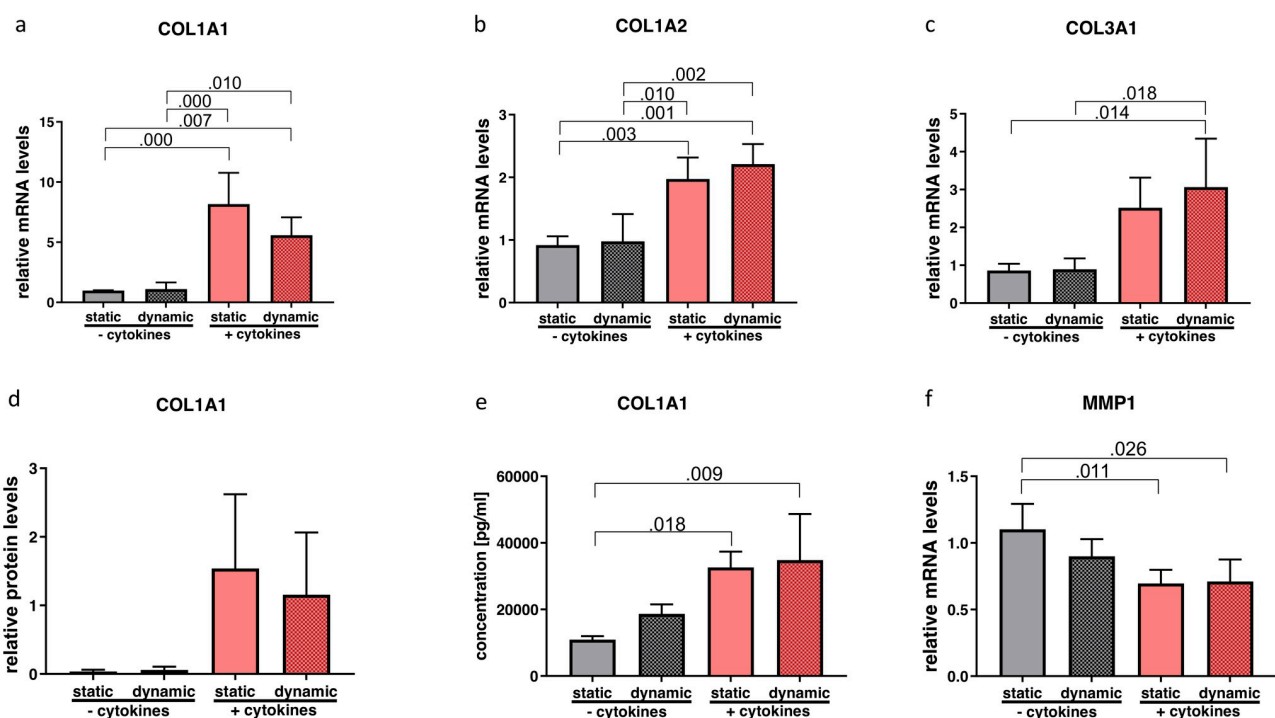

**Fig 4. Effect of cytokine treatment and/or vibration on ECM-related molecules.** mRNA (a-c and f) and protein levels (d and e) of ECM-related molecules were analyzed by RT-qPCR, Western blot and Luminex, respectively. Results are represented as mean ± S.D. of four independent experiments (N = 4). Statistical analysis was performed using one-way ANOVA (normally distributed data) or Kruskal-Wallis test (non-parametric data), $p < 0.05$ was considered significant; decimal numbers above the bars represent the statistically significant p-values from post-hoc comparisons. For the p-values not presented here, see S2 File. *COL1A1* (collagen 1 alpha 1), *COL1A2* (collagen 1 alpha 2), *COL3A1* (collagen 3 alpha 1), *MMP1* (matrix metalloproteinase 1).

### *In vivo* studies

The benefit of voice rest following an injury remains a subject of debate, since a considerable number of factors contribute to the healing process leading to inconclusive results [17, 57]. Kaneko et al. investigated the role of voice rest based on vocal and laryngoscopic/stroboscopic parameters. They assigned patients who underwent phonosurgery to two groups. Participants had to comply with voice rest for three or seven days followed by voice therapy, with follow-up examinations at 1, 3 and 6 months postoperatively. Subjective parameters (GRBAS and VHI-10), as well as objective parameters (normalized mucosal wave amplitude, jitter and shimmer) improved significantly in the 3-day group but not consistently and not at all time points. Nevertheless, the authors concluded that early phonation could lead to favorable wound healing and functional recovery [18].

To date there are no studies exploring the molecular consequences following voice rest or voice load after phonosurgery, and only a handful of studies have investigated the effects of vocal overload on the molecular level in humans. Verdolini et al. analyzed laryngeal secretions of three vocally healthy participants undergoing vocal loading followed by voice rest. They described a decrease of inflammatory markers in participants who followed resonant voice exercises compared to those who observed voice rest. In their setting, laryngeal secretions were gathered from topically anesthetized VF in a very complex manner by using cotton swabs, and as a consequence, only one patient from each group completed the study protocol [15].

There are several publications investigating the consequences of VF injury in animals, mostly rats. They reported an altered expression of certain inflammatory and pro-fibrotic

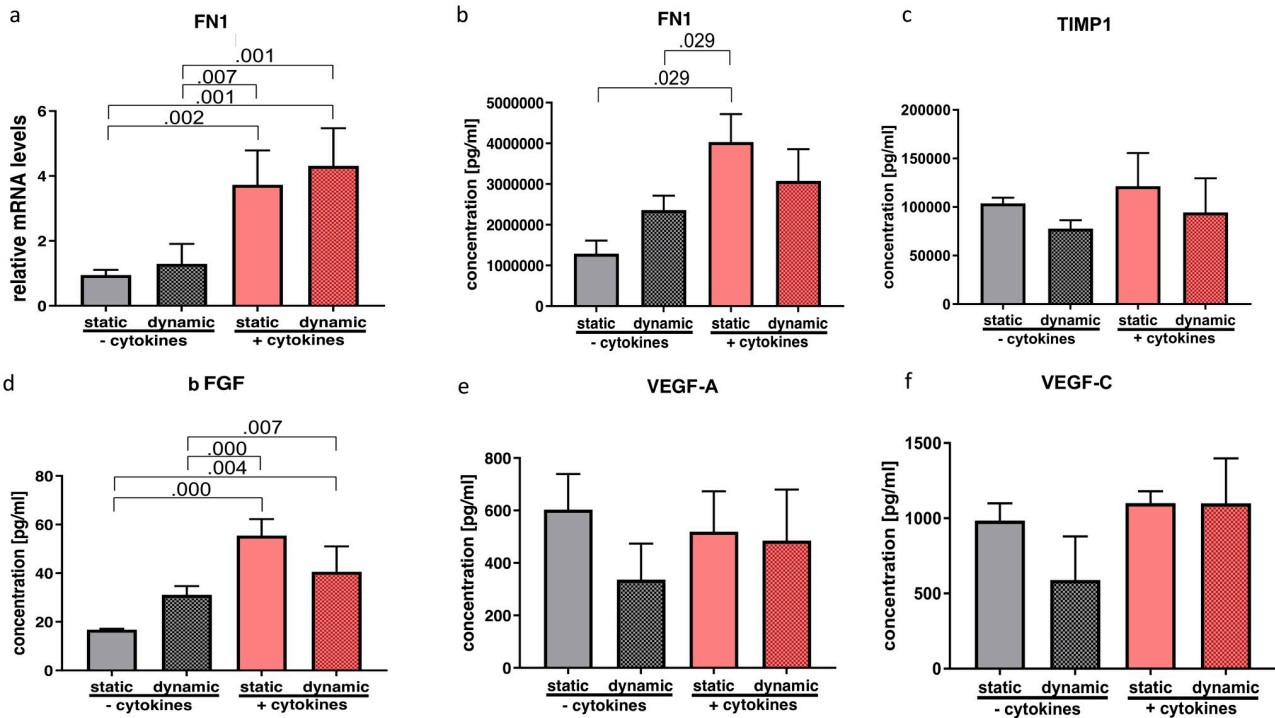

**Fig 5. Effect of cytokine treatment and/or vibration on ECM-related molecules, growth factors and angiogenic factors.** mRNA (a) and protein levels (b-d) of ECM-related molecules as well as growth- and angiogenic factors were analyzed by RT-qPCR and Luminex, respectively. Results are represented as mean ± S.D. of four independent experiments (N = 4). Statistical analysis was performed using one-way ANOVA (normally distributed data) or Kruskal-Wallis test (non-parametric data), p<0.05 was considered significant; decimal numbers above the bars represent the statistically significant p-values from post-hoc comparisons. For the p-values not presented here, see S2 File. *FN1* (fibronectin 1), *TIMP1* (Tissue inhibitor of metalloproteinase), bFGF (basic fibroblast growth factor), VEGF A (vascular endothelial growth factor A), VEGF C (vascular endothelial growth factor C).

factors (*IL1β* and *TGF-β1*) or several ECM-related genes (*HA, collagen*) [6, 58]. In an *in vivo* rabbit model, Rousseau et al. investigated wound healing processes after an acute phonotrauma, focusing on the epithelial layer of the VF. Again a significant upregulation of *IL1β* and *TGF-β1* was revealed after 120 minutes of phonatory stress [59]. In line with these results, the upregulation of ECM- and inflammation-related genes and proteins in our experiments, such as collagen, HA as well as *IL1β, TGF-β1* and *IL6* reflect the applicability of our model, even though only one cell type was included.

### *In vitro* studies

To our knowledge, only one publication has explored the impact of mechanical stimulation on VFF under inflammatory conditions to date. Branski et al. exposed VFF of rabbits to cyclic tensile strain in the presence or absence of IL1β [2]. They described a reduced expression of the inflammatory mediators *iNOS* and *COX2* as well as *MMP1*. They concluded that dynamic stimulation at appropriate amplitudes might have an anti-inflammatory effect, which is in accordance with our results. However, their device could only apply a very limited range of frequencies, whereas our device covers the entire range of the human voice [29].

Another problem of *in vitro* experiments is the aqueous environment surrounding the cells. *In vivo*, cells are embedded in a highly crowded environment with plenty of macromolecules known to have equally important kinetic and thermodynamic functions such as setting the

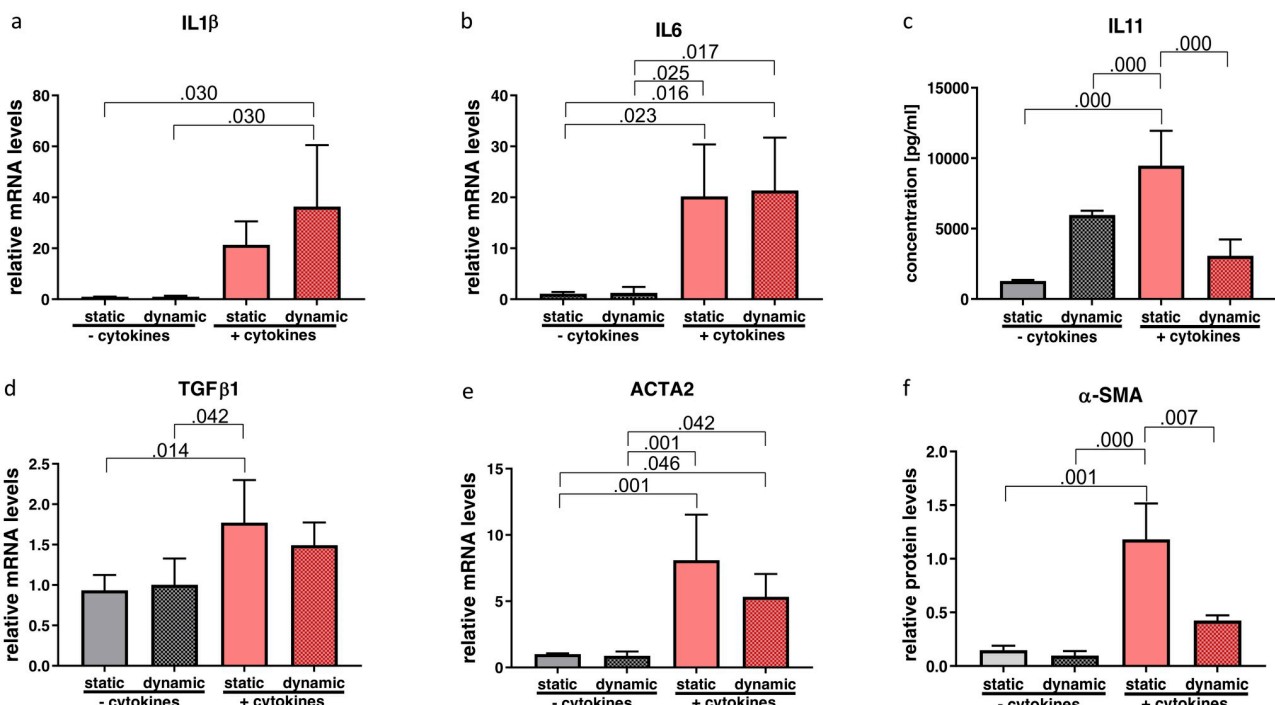

**Fig 6. Effect of cytokine treatment and/or vibration on inflammatory and fibrogenic markers.** mRNA (a, b, d, e) and protein levels (c and f) of inflammatory and fibrogenic markers were analyzed by RT-qPCR, Western blot and Luminex, respectively. Results are represented as mean ± S.D. of four independent experiments (N = 4). Statistical analysis was performed using one-way ANOVA (normally distributed data) or Kruskal-Wallis test (non-parametric data), $p < 0.05$ was considered significant; decimal numbers above the bars represent the statistically significant p-values from post-hoc comparisons. For the p-values not presented here, see S2 File. *IL1β* (interleukin 1β), *IL6*, IL11, *TGFβ1* (transforming growth factor beta 1), *ACTA2* (alpha smooth muscle actin), α-SMA (alpha smooth muscle actin).

optimal pH. Graupp et al. demonstrated in a VFF model that macromolecular crowding in combination with TGF-β achieves a significantly higher concentration of ECM components (e.g. collagen and HA) as well as of α-SMA compared to an uncrowded control [35, 36]. For this reason the experiments presented here were performed under crowded conditions. A direct comparison with other studies which disregard macromolecular crowding is not possible.

As mentioned earlier, VFF were exposed to a serum free medium for 24 hours. It is known that starvation influences the susceptibility of cells to inflammatory stimuli. On the other hand, it is a well-established method for making cell populations more homogeneous and improving reproducibility. This method has already been applied in previous VF fibrosis models [43, 60, 61].

### Effects of vibration and cytokine-stimuli on VFF

Prior to this study, we investigated the effects of IL1β and TGF-β1 separately as well as in combination, at different points in time, on the expression of other cytokines and ECM-related proteins (see S1 File and S1 Fig). These two cytokines and their concentration were selected based on previous publications on inflammation *in vitro* [2, 6, 58, 62–65]. As a consequence of our pretrials, the combination of IL1β and TGF-β1 was used for the present investigation, as both cytokines are known to play important roles in acute inflammatory processes and tissue fibrosis.

In the current study, no deteriorating effects of vibration were observed on two of the most important VF ECM components, namely HA and collagen. Gene expression of *HAS1*, *HAS3*, as well as HA concentrations in supernatants were significantly increased following an inflammatory and pro-fibrotic stimulation. Importantly, vibration did not diminish HA levels, which is significant since this glycosaminoglycan has important anti-fibrotic properties. Some other studies on the contrary showed changes in the HA metabolism following mechanical stimulation [25, 29]. This might be due to differences in stimulatory patterns and their duration, since it is known that VF tissue responds differently depending on the type of phonatory stimulus [66].

In addition, we demonstrated an increase in *COL1A1*, *COL1A2*, *COL3A1* gene expression and extracellular collagen I concentration following inflammatory stimuli, which is again in line with previously mentioned animal experiments [6, 58].

Consistently with a previous review, *FN1* gene expression and its encoded protein, fibronectin, were significantly upregulated under cytokine exposure on gene as well as on protein level, whereas vibration had no effect [67].

We did not see any changes of other cytokines typically described in early VF injury rat models such as COX2 [6, 58]. This might be explained by the fact that in the present *in vitro* study only hVFF were used and therefore other cell types such as macrophages could not be taken into consideration. There are some limitations in the present study. Firstly, even though the bioreactor is able to expose the cells to a wide range of frequencies and to activate some mechanotransductive pathways, we are aware that the mechanical forces during phonation are much more complex and not reproduced to their full extent here. Secondly, only one cell type was used, whereas inflammatory reactions *in vivo* involve multiple cell types. Particularly, the interaction of VFF and macrophages seem to have a huge impact on the inflammatory response [68]. And thirdly, the fact that the VFF are immortalized might lead to different behavior compared to near primary cells or to *in vivo* cells.

Our *in vitro* experiments revealed a significant decrease of protein expression of α-SMA following four hours of vibration per day compared to the static group. The gene expression level of *ACTA2*, the gene encoding for α-SMA, also decreased, but without reaching statistical significance. It is known that VFF transformation into myofibroblasts during inflammation is accompanied by an increase in α-SMA, a well-established myofibroblast marker [29, 69, 70]. Myofibroblasts appear to be responsible for wound contraction as well as increased collagen deposition and could lead to a deterioration of VF oscillation [70]. Our results showed reduced transformation into undesired myofibroblasts, reflecting beneficial effects of vibration.

In parallel, we observed a significant decrease of IL11 in supernatants, when vibration was applied. IL11 belongs to the IL6-family. It seems to play a pivotal role in the down-streaming pathway of TGF-β1 and therefore represents an essential component of fibrogenesis, such as in cardiovascular fibrosis. Another recent publication indicates that IL-11 is also a key factor in the development of multiple sclerosis. Consequently, it might be a potential therapeutic target to avoid fibrosis [71–73]. Both α-SMA and IL-11 are known to be involved in the extracellular-signal-regulated-kinase signaling pathway, which plays a role in mechanotransduction. Previous studies revealed a connection between mechanical stimulation and anti-inflammatory effects [71, 74].

## Conclusion

The present study aimed to examine the impact of mechanical stimulation on human VFF under inflammatory and pro-fibrotic conditions for the first time. Our experiments revealed a significant decrease of fibrosis-linked proteins IL11 and α-SMA following mechanical

stimulation. Notwithstanding the inherent limitations of an *in vitro* study, it might suggest that a certain amount of postoperative VF vibration might have a beneficial impact on wound healing. Further research should focus on the approximation of *in vivo* inflammatory conditions by for example co-cultivating VFF with other cell types and developing appropriate three-dimensional models.

## Supporting information

**S1 Fig. Changes on gene expression due to inflammatory stimuli over time.** This supplementary figure shows the effect of IL1β, TGFβ1 and their combination on the expression of ECM-related proteins and cytokines over time compared to a non-treated control group. *HAS1* (hyaluronan synthase 1), *HAS2* (hyaluronan synthase 2), *HAS3* (hyaluronan synthase 3), *COL1A1* (collagen 1 alpha 1), *IL6* (interleukin 6), *TGFβ1* (transforming growth factor beta 1). (TIF)

**S1 File. Explanation of the pre-trials methodology.** This supplementary file describes in detail the procedure in our previously performed experiments. The presented cytokines in the manuscript were based on these results. (DOCX)

**S2 File. Excel spreadsheet containing raw data from statistical tests.** Each sheet contains the individual data of qPCR, Luminex, Western Blot and ELISA. (XLSX)

**S3 File. PDF sheet with catalogue numbers.** (PDF)

**S4 File. Excel spreadsheet containing standard curves of each Luminex sample.** (XLSX)

## Acknowledgments

Doctoral school Molecular Medicine and Inflammation, Medical University of Graz.

## Author Contributions

**Conceptualization:** David Hortobagyi, Markus Gugatschka.

**Data curation:** David Hortobagyi, Tanja Grossmann, Magdalena Tschernitz, Magdalena Grill.

**Formal analysis:** David Hortobagyi.

**Investigation:** David Hortobagyi, Tanja Grossmann, Magdalena Tschernitz, Magdalena Grill, Andrijana Kirsch.

**Methodology:** David Hortobagyi, Tanja Grossmann, Magdalena Tschernitz, Magdalena Grill, Andrijana Kirsch, Claus Gerstenberger, Markus Gugatschka.

**Project administration:** David Hortobagyi, Markus Gugatschka.

**Resources:** David Hortobagyi, Claus Gerstenberger, Markus Gugatschka.

**Supervision:** Markus Gugatschka.

**Validation:** David Hortobagyi, Markus Gugatschka.

**Visualization:** David Hortobagyi, Markus Gugatschka.

**Writing – original draft:** David Hortobagyi, Tanja Grossmann, Markus Gugatschka.

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
