## [Decision Letter · Decision Letter 0]

13 Jul 2020

PONE-D-20-19358

Voice rest following vocal fold injury: The molecular perspective

PLOS ONE

Dear Dr. Hortobagyi,

Thank you for submitting your manuscript to PLOS ONE. After careful consideration, we feel that it has merit but does not fully meet PLOS ONE’s publication criteria as it currently stands. Therefore, we invite you to submit a revised version of the manuscript that addresses the points raised during the review process.

We look forward to receiving your revised manuscript.

Kind regards,

Marie Jetté

Academic Editor

PLOS ONE

Journal Requirements:

Reviewers' comments:

Reviewer's Responses to Questions

**Comments to the Author**

1. Is the manuscript technically sound, and do the data support the conclusions?

Reviewer #1: Partly

Reviewer #2: Yes

Reviewer #3: Yes

Reviewer #4: Partly

2. Has the statistical analysis been performed appropriately and rigorously? 

Reviewer #1: No

Reviewer #2: Yes

Reviewer #3: No

Reviewer #4: Yes

3. Have the authors made all data underlying the findings in their manuscript fully available?

Reviewer #1: No

Reviewer #2: No

Reviewer #3: No

Reviewer #4: No

4. Is the manuscript presented in an intelligible fashion and written in standard English?

Reviewer #1: No

Reviewer #2: No

Reviewer #3: Yes

Reviewer #4: Yes

5. Review Comments to the Author

Reviewer #1: The paper used a custom bioreactor to simulate vibratory stresses as of human phonation. The goal of this study was to evaluate short-term effects of vibration on human vocal fold fibroblasts primed with IL1-beta and TGF-beta1. The overall motivation of the study was clinically relevant. However, major concerns especially on the methodology and results sections will need to be addressed. Specific comments are as below.

(1) Introduction: The literature review is insufficient. A more thorough review on existing vocal fold bioreactors (in vitro work) such as Barlett et al., 2019, Kim et al., 2016 and Latifi et al., 2016 as well as the perspective of voice rest after phonosurgery (in vivo work) such as Dhaliwal et al., 2019, Whitling et al., 2018 and Rinkanen et al., 2019 should be reviewed descriptively.

(2) Methodology: In general, more details about the protocols are expected for experimental replication. Please include catalogue numbers for all biochemical/ molecular assays, e.g., LDH Assay, lysis reagent etc.

(3) Cell culture and treatment: Please describe which cell passages were used. For the BioFlex culture plates, were the cells seeded on a plastic membrane? If so, what was the pore size and other structural/ membrane properties of the member? All these factors will affect cell adhesion, diffusion, mobility and proliferation. Also, please provide more details what exactly "inert macromolecules" were used and why crowded conditions were constituted and required. Would the presence of the macromolecules confound the results? Explain why IL-1beta and TGF-beta1 and at such dose/ exposure were used to prime the cells. Any validation tests were done to ensure that the cytokine priming worked. If so, please present the results.

(4) Mechanical stimulation: More details about the oscillation such as vibratory amplitude are needed to provide a better view of the bioreactor functionality.

(5) LDH assay: This paragraph needs more editing as some punctuation marks or grammatical errors were spot. Also, what's the volume of supernatant collected for the assay?

(6) RNA isolation: Any estimation of the cell retrieval efficiency for the RT-qPCR? If the cells were strongly attached on the membrane, it would not be easy to retrieve all cells and the harsh reagents may change the phenotype of the cells. The selection of gene markers should be justified.

(7) Western blot analysis: For the primary antibodies, were they mono- or polyclonal? The selection of intracellular protein markers should be justified.

(8) Luminex: Please provide more details about the procedures and the protocol, e.g., standards, blank controls etc. The selection of the protein markers should be justified. Usually when multiplex assay was run, a validation step of selected markers are needed. Please explain if any validation was done with the Luminex results. If not, why not?

(9) ELISA: Please provide more details about the procedures and the protocol, e.g., blank controls, catalogue number etc. The selection of the single HA marker should be justified.

(10) Statistics: Please make it clear which variables were treated as "between-subject" independent variables and which comparisons were made in the design.

(11) Results: Please report all results include those of LDH and the TIMP1. Also, the presentation of the results were mixed with PCR, mRNA, Luminex and ELISA. As the choice of markers lacks justification in the methodology, it made the results herein very difficult to follow and comprehend. A reorganization of this section in terms of writing and figures (Figure 2-5) is strongly needed. Please include asterisk for significant values. For the decimal numbers above the bars (Figure 2-5), what did these numbers represent? Were they p-value from ANOVA or post-hoc pairwise comparison?

(12) Results: Figure 3. For Col1A1, the pattern of changes for "+ cytokine" between static and dynamic conditions are different for (d) and (e). Why so? Such results should be in sort of agreement.

(13) Discussion: In general, please denote clearly which figure that the results were referred to in the main text. Also, some claims were not supported by the results. For example, Line 321-324 claimed that their results showed up-regulated ECM- and inflammation-related genes and proteins. However, Figure 5 showed the downregulation but not the upregulation for ACTA2, TGFbeta 1, IL-11 and alpha-SMA under the "+ cytokine/ dynamic" condition, compared to the "+cytokine/ static" condition.

(14) Please include a more thorough discussion around the biological/ physiological functions of IL-11 on fibroblasts and the synergistic interactions of IL-11 with other cytokines.

(15) Please discuss what the limitations and future direction of this study.

Reviewer #2: The main finding in this paper is that vibration of hVFFs results in reduced expression of pro-inflammatory cytokines and smooth muscle actin while in an induced inflammatory state.The ECM component HA which is necessary for reduction of fibrosis is not influenced by vibration.

The introduction is brief and fails to acknowledge clinical studies addressing vocal rest/use post-surgery, findings which provide necessary context to this manuscript. The methodology is well designed and reflects an appropriate number of replicates. The results are of interest to the field and align with early ambulatory recovery guidelines in other disciplines. The discussion section fails to address any limitations of the study and is oversimplified such that some statements are partially incorrect. If the findings of this study translate to human voice they may have a significant impact on post-surgery guidance for voice use following phonosurgery.

The manuscript would benefit from external copy-editing as there are many instances where the meaning is unclear, or wording is inappropriate. No supplementary materials were available in the reviewer’s download so it is unclear if all the data is freely available.

Specific comments:

Title: gives the impression this is a review or conducted in humans, should reflect study more accurately.

Abstract:

Line 18: word choice - Reword to reflect that biopsies for molecular biology studies are unethical, as biopsies are possible.

Line 19: word choice - As the model had been established prior (cell line Thiebault, biomechanical reactor in a previous publication) reword to reflect utilization of a previously developed in vitro phonomimetic biomechanical model.

Line 24: word choice - Wording ambiguous as to whether the mechanical stimulus or cytokine stimulus is applied for 4 hours.

Introduction:

This section fails to address the role of voice therapy in the treatment of vocal fold lesions – for most lesions voice therapy is recommended in combination with surgery, and in some cases alone. Omission of voice therapy weakens this section substantially.

Line 41: word choice - sentence meaning is unclear. Reword to reflect damage caused surgically, or mechanically by phonotrauma lead to an upregulation of inflammatory cytokines.

Line 44: word choice - use of “the worst-case” unnecessary and changes sentence meaning.

Line 46/47: word choice – reword to reflect that these findings may influence clinical practice guidelines.

Line 57: The statement, “barely any sound evidence,” is inappropriate. There are several peer reviewed articles addressing voice rest/use following phonomicrosurgery. See Bercirovic et al. 2019, Whitling et al. 2018, Kiagadaki et al. 2014, and Wang and Huang 1994. The findings of these studies lack consensus and serve to strengthen the authors arguments.

Line 59: word choice – the healing process is influenced/impacted by these factors, “trigger” is not correct in this usage.

Line 65: word choice – consider necessary or essential as opposed to mandatory, changes sentence meaning.

Line 66/67: ECM production changes are inadequately referenced.

Line 68: While there is not much evidence, Zhang et al. 2015 describe the impact of cyclic tensile strength on hVFFs in the presence of cigarette smoke and should be included here.

Methods:

Line 85: word choice – remove “Therefore.”

Line 94: word choice – sentence does not make sense. Reword to reflect that inclusion of inert macromolecules enhances ECM production such that it is easier for researchers to evaluate them in culture.

Line 99: word choice – “foreseen” used incorrectly, consider stratified/randomized/ or assigned to throughout the manuscript

Line 119: word choice – reads awkwardly. Consider rewording “orientating on the publication of…” to, “as described by.”

Line 119: double period at end of sentence.

Line 120: word choice – “foreseen” incorrectly used

Line 139/140: How was the RNA quality established? No mention of RIN or the use of a denaturing agarose gel. This data is essential to ensuring the RNA meets quality standards for qPCR.

Line 144: Please include more details for the reaction (number of cycles, temperature, nucleic acid dye, etc.) Looking at the paper referenced in this section this too lacks details about the qPCR methods.

Line 152: double period at end of the sentence.

Statistical Analysis:

General: Include effect sizes in the manuscript as these may demonstrate clinical implications where non-significant “trends” are observed. Consider Gaeta and Brydges 2020 JSLHR article when reporting effect size.

Results:

General: Include p values and effect sizes in the text describing the results.

General: Include p values and effect sizes in the Figure legends.

General: This section would benefit from external copy-editing. A number of sentences are unclear in their meaning, and there is a repeated use of some phrases which read unclearly or strangely, e.g. failed statistical significance, not altered by.

Line 213: word choice – change “under” to with.

Line 214: word choice – consider rewording to, “Gene expression results were consistent with significantly elevated transcript levels of HAS1 and HAS3.”

Line 231: word choice – reword “failed statistical significance” to something like “did not meet statistical significance” throughout.

Line 246: word choice – insert, “and was” before, “not altered by additional…”

Line 262: Figure legend for Fig 4 is incomplete.

Line 265 – 267: word choice – sentence meaning is unclear. Do the authors mean that non-significant upregulation was observed with cytokine treatment alone?

Line 274: word choice – reword, “under” to, “in the.”

Discussion:

General: There is no discussion of limitations within the study. These include:

1) The use of SYBR for qPCR. SYBR dye binds to double stranded DNA in a non-specific manner, thus does not confirm that the correct transcript has been replicated. Use of a Dual-labeled Primer (e.g. TaqMan) is more specific and reproducible.

2) Immortalized cell lines lack a bloody supply, and thus inflammatory observations are made outside of a normal physiologic system and may not be translatable to in vivo.

3) Immortalized cell lines have inherent property changes from primary cells or in vivo cells and may behave differently.

4) In vitro biomechanical manipulation is an approximation of voice and may not apply identical shearing and stress forces as observed in physiologic voice

Line 292: The role of inflammation is grossly understated. Inflammation is also required for homeostasis/adaptation to stress/promotion of healing/recruitment of host immune response, etc.

Line 319: Italicize in vivo.

Line 326: Statement is incorrect. Rousseau model is a controlled vibratory phonation model using electrical stimulation to elicit vocalization. See Kimball et al. 2019 for information on stimulation and vibratory patterns in this model.

Line 340: spelling incorrect – glycosaminoglycan

Line 346: word choice – use of, “could” is ambiguous, consider, “demonstrated an increase…”

Line 352: word choice – use of, “could” is ambiguous, consider, “did not…”

Line 354: Please change ex vivo to in vitro. Ex vivo implies minimal changes to the tissue upon removing it from the body.

Line 387: italicize in vivo

Figures:

General: Consider removing the p values from the figures and use Asterix to denote the level of significance. The p values make the graphs seem over complicated.

Reviewer #3: This is an interesting paper describing application of a sophisticated bioreactor for studying inflammatory signaling in VFFs in concert with a vibratory stimulus. The bioreactor technology has previously been published by the group. This work is certainly a useful contribution to the literature, and I offer the following comments and suggestions:

1. Is the vibratory dose delivered to the 6 well plate uniform across all wells, or does it vary based on the position of the well? If there is well-to-well variation, how significant is this and how was it managed?

2. MMC was used in all conditions to enhance biological responsiveness and more closely simulate a fibrosis situation, however this is just mentioned briefly in the Method section. It would be very helpful to show the effect of MMC on the cytokine and vibration responses here – ideally with a non-MMC experimental condition or, if not practical, by describing the results in light of this groups’ prior publication that showed the influence of MMC alone on VFF biology in vitro. This really should have more emphasis, especially when the authors interpret and compare their results to those of others who did not use MMC in their experiments.

3. I appreciate the care given to Tukey adjustment and parametric versus non-parametric statistical test selection, but wonder if these data would be best handled by a 2-way ANOVA in which cytokine treatment and vibration treatment are handled as independent variables, with their interaction effect included. The interaction between these experimental stimuli may be important here. Also, with no word or page limits in this online journal, I please report the results of the F and KW tests (not just the significant pairwise comparisons), as well as the nonsignificant LDH data. These can easily be placed in a supplementary materials section.

4. Finally, I suggest more careful and conservative description of the clinical and wound healing issues that motivate the work and are described in the Introduction and Discussion. For example:

a. Early inflammation is not just for avoiding infection (start of Discussion). There is much more complex biology involved.

b. I disagree that there is a lack of clinical recommendations for voice rest (Abstract); rather there is a lack of UNIFORM recommendations.

c. The comparison of VF vibration with orthopedic mobilization after injury or surgery is a more complex and nuanced comparison than is suggested (Introduction). These are quite different mechanical situations and I suggest describing in more detail.

d. I suggest citations to support “size of the VF lesion, smoking status, sex, age etc” (Introduction).

e. Paragraph 2 in the Discussion might fit better in the Introduction.

f. I think that “in vivo” (line 387, Conclusion) was intended to be “in vitro”.

Reviewer #4: The purpose of the current study was to explore the effects of vibration on hVFF in an inflammatory and normal state. I very much like the premise of this study as it provides an important avenue to discuss biological outcomes associated with mechanical stimulation following inflammation. However, I have concerns regarding manuscript overall clarity, methodological decisions, and the conclusion. Specific comments are listed by section.

Title / Abstract

• I would consider changing the title of the manuscript. First, it sounds like the paper is a Review as opposed to an experimental study. Also, it is truly not reflective of what was done in the study. Vocal folds were not injured. VFF were treated with an inflammatory challenge.

• The Introduction statement of the abstract is not clear. The important role of voice rest is introduced, but then stated that the authors seek to evaluate the effect of vibration on VFF.

Introduction

• Not all readers will be familiar with voice rest recommendations following phonosurgery. The Introduction would benefit from a brief overview (1-2 sentences) of typical voice rest recommendations. This would provide some context for the reader.

• The authors begin the Introduction discussing upregulation of inflammatory cytokines following phonotrauma, but not mention of the inflammatory cascade following phonosurgery. This should be mentioned as I think the authors are making of case to study the role of voice rest following phonotrauma and/or vocal fold surgery.

• The Introduction needs further information (1-2 sentences) regarding the important role of VFF in vocal fold physiology.

• In general, it is unclear that the authors are trying to mimic inflammation following phonotrauma or phonosurgery and the subsequent role of vibration versus rest.

• The objective of the study is listed as studying the cellular responses of hVFF following a profibrotic and inflammatory stimulus under static and dynamic conditions. I have several concerns with this statement. First, cellular responses is extremely broad and can encompass a huge range of factors. I would recommend expanding upon this. Furthermore, the authors state in the title that they are seeking to study the molecular, not cellular perspective. It is unclear that static and dynamic is supposed to related to vocal fold rest and vibration, respectively. Finally, while inflammation if briefly introduced, what is the importance of a pro-fibrotic stimuli?

Methods

• Please provide justification regarding the choice to serum starve the VFF. Serum starvation has been shown to cause cells to undergo apoptosis and increase cell susceptibility to inflammatory stimuli.

• The term pro-fibrotic and inflammatory culture conditions is utilized. While the use of the inflammatory makes sense, pro-fibrotic is confusing. Please provide justification / citations regarding the pro-fibrotic effect of these cytokines.

• It is stated that these cytokines were capable of maintaining a consistent inflammatory reaction for at least 72 hours. How was this measured / verified?

• I realize that the cells were only vibrated for four hours, but in the initial section regarding cell culture and treatment it reads like the cells were vibrated for 72 hours.

• In Mechanical Stimulation, I know that a citation is provided, but it would still benefit from 1-2 sentences justifying how vibration parameters and time relate to human voice production.

• It is unclear to me how the cell viability assay was conducted. The authors state to evaluate maximum LDH activity, cells were seeded in parallel….. Were the cells seeded after mechanical stimulation for the purpose of this assay? How would that potentially affect the viability results? Please clarify.

• In addition, the authors state that LDH activity of the samples was expressed as percentage of the maximal LDH activity. However, this data is never presented.

• There is no justification in the Introduction or Methods regarding the choice to evaluate various classes of genes and proteins. It is not until the Discussion that the importance of looking at these factors is even mentioned, albeit relatively briefly. This should really come earlier in the manuscript and be expanded.

Results

• Although NS, I feel the manuscript would benefit from inclusion of the viability data.

• Subheading including ECM, angiogenic factors, fibrogenic markers are first introduced in the Results section. These are important classes of genes and proteins and it is strange to see these subheadings for the first time in the Results section.

• I find the results very difficult to follow. Specifically, the most important findings (e.g. IL11) are difficult to realize, because so many genes are discussed typically individually. I believe there some redundancies were the same genes demonstrate the same findings in terms of significance related to inflammatory condition and vibration. I wonder if there is a way for the authors to condense / re-organize.

• The authors frequently use the terms trending towards significance, near significance etc. I would suggest being more judicious with the use of these terms. These are not significant findings. Any interesting trends the authors may wish to save for the Discussion.

Discussion

• The overall setup for the manuscript is much clearer in the first paragraph of the Discussion than any other part of the manuscript.

• The authors indicate that different inflammatory and pro-fibrotic stimuli were investigated based on previous publications of in vitro inflammation. Were these studies with VFF or other cell types? Furthermore, on page 17 the authors discuss inflammatory cytokines following vocal fold injury in vivo. This Discussion should be moved up and used to justify choice of inflammatory challenge.

• Why is there not mention of the viability data? It is interesting to me that cytokine treatment or vibration did not affect cell viability.

• IL1B was significantly increased in the inflammatory dynamic condition. Why was this finding not discussed?

• The authors state in the current study that no deteriorating effects of vibration were observed on two important ECM components. Were deleterious effects expected? The vibration parameters were not supposed to be traumatic, correct?

• In the conclusion, the authors state “Notwithstanding the inherent limitations of an in vivo study, it might suggest that a certain amount of postoperative/-traumatic vocal load might have beneficial impact on wound healing.” First, this is not an in vivo study. Second, are the authors suggesting that traumatic vocal load might have a beneficial impact on wound healing. There is a significant difference between traumatic vocal load and likely restorative vocal fold vibration. I do not feel comfortable or support the statement that traumatic load is good for wound healing. Was the mechanical stimulation in the current study meant to mimic a traumatic load? This statement needs to be clarified.

Figure / Figure Captions / Other

• In every Figure caption the statistical analysis methods are stated. This is redundant and in the manuscript text. I suggest removing.

• Please proofread carefully. Numerous typos and inconsistencies are appreciated throughout the manuscript.

6. PLOS authors have the option to publish the peer review history of their article (what does this mean?). If published, this will include your full peer review and any attached files.

Reviewer #1: No

Reviewer #2: No

Reviewer #3: No

Reviewer #4: No

---

## [Author Response · Author response to Decision Letter 0]

23 Aug 2020

Reviewer #1

Comment #1

Introduction: The literature review is insufficient. A more thorough review on existing vocal fold bioreactors (in vitro work) such as, Kim et al., 2016 and Latifi et al., 2016 as well as the perspective of voice rest after phonosurgery (in vivo work) such as Dhaliwal et al., 2019, Whitling et al., 2018 and Rinkanen et al., 2019 should be reviewed descriptively.

Answer #1 

Aspects of other bioreactors and in vivo works were supplemented. 

The following modification was made based on comment #1 in manuscript, introduction, lines 86-99: For this purpose, various bioreactors, trying to imitate phonatory stresses in vitro, were developed in the past years. Farran et al. and Kim et al. built bioreactors where cells were seeded onto flexible membranes which were then set into vibration either by an air stream or by linear actuators, respectively. Since the cells in these models are single-layered, these devices are described as ‘two-dimensional’ reactor types. 

Different approaches were used by Gaston et al. and Titze et al. who developed three-dimensional scaffolds where cells are predominantly exposed to tensile stresses. 

Recently, the bioreactor which is closest to the in vivo situation was developed by Latifi et al.. They injected a scaffold, consisting of human VF fibroblasts (hVFF), into silicone VF replicas. Cells were exposed to mechanical forces by applying an airflow from beneath which creates a similar mode of vibration to that of the human VF in the larynx. 

Comment #2

Methodology: In general, more details about the protocols are expected for experimental replication. Please include catalogue numbers for all biochemical/ molecular assays, e.g., LDH Assay, lysis reagent etc.

Answer #2

Materials and methods were expanded in order to get a more detailed understanding of the working steps that we performed. Catalogue numbers of the products we used were added to the supplementary materials. Since the changes to materials and methods were considerable, we refrain from inserting all amendments to this reply. However, all changes are labelled in the text in red.

Comment #3

Cell culture and treatment: Please describe which cell passages were used. For the BioFlex culture plates, were the cells seeded on a plastic membrane? If so, what was the pore size and other structural/ membrane properties of the member? All these factors will affect cell adhesion, diffusion, mobility and proliferation. 

Answer #3 

The following modifications were made based on comment #3 in manuscript, materials and methods, lines 122-123: The mean number of cell passages of the four experiments was 125.

The following modifications were made based on comment #3 in manuscript, materials and methods, lines 125-128: The Bioflex silicone membranes were coated with ProNectin, a fibronectin-like engineered protein. The coating was selected during previous projects, as cell morphology was identical to standard culture conditions. 

Comment #4

Also, please provide more details what exactly "inert macromolecules" were used and why crowded conditions were constituted and required. Would the presence of the macromolecules confound the results? 

Answer #4

Thank you for this comment. The details on the inert macromolecules has been described in materials and methods, lines 142-144.

Furthermore, the importance of macromolecular crowding was added in the discussion, lines 530-538: Another problem of in vitro experiments is the aqueous environment surrounding the cells. In vivo, cells are embedded in a highly crowded environment with plenty of macromolecules known to have equally important kinetic and thermodynamic functions such as setting the optimal pH. Graupp et al. demonstrated in a VFF model that macromolecular crowding in combination with TGF-� achieves a significantly higher concentration of ECM components (e.g. collagen and HA) as well as of �-SMA compared to an uncrowded control. For this reason the experiments presented here were performed under crowded conditions. A direct comparison with other studies which disregard macromolecular crowding is not possible.

Since the in vivo microenvironment of cells is also “crowded”, we assume that the presence of macromolecules is rather an approximation to the “real” world than a confounding factor. 

Comment #5

Explain why IL-1beta and TGF-beta1 and at such dose/ exposure were used to prime the cells. Any validation tests were done to ensure that the cytokine priming worked. If so, please present the results. Supplementary materials

Answer #5

The dose used was based on previously published experiments and has been written in materials and methods lines 132-139. A more detailed explanation of our previously performed experiments was added in the supplementary appendix. 

The following modifications were made based on comment #5 in manuscript, materials and methods, lines 135-139: This combination and these concentrations of cytokines were used based on previous publications and preliminary trials by our group, in which cytokines were capable of causing durable changes in the expression of other cytokines and ECM-related proteins for at least 72 hours (see supplementary appendix).

Comment #6

Mechanical stimulation: More details about the oscillation such as vibratory amplitude are needed to provide a better view of the bioreactor functionality. 

Answer #6

Thank you for this valuable comment.

The following modifications were made based on comment #6 in manuscript, materials and methods, lines 210-220: The stimulatory pattern, a linear sinusoidal chirp sound in a frequency range of 50 Hz to 250 Hz corresponds to the frequency range of the primary larynx sound in humans. The membrane displacement in the center of a well was dependent on the applied frequency and averaged 82 µm ± 5 µm. The input voltage (V) on the loudspeaker was set to 1.1 V and served as a measure of the vibration intensity. Stimulation was applied for four hours per day, following the publication of Titze et al. There are a handful of publications dealing with displacements (vertical and horizontal) of human vocal folds during phonation, reporting a wide range of values (depending on the setting and/or measurement method). The simulation of these displacements is reproducible with our bioreactor and complies with the maximum displacement in humans. 

Comment #7

LDH assay: This paragraph needs more editing as some punctuation marks or grammatical errors were spot. Also, what's the volume of supernatant collected for the assay? 

Answer #7

Thank you for this suggestion. Regarding punctuation marks and grammatical errors and grammatical errors, an external copy editing was performed. 

The following modifications were made based on comment #7 in manuscript, materials and methods, lines 232-233: Subsequently, 50µL supernatants of all wells were collected and processed in duplicates according to the manufacturer’s instructions.

Comment #8

RNA isolation: Any estimation of the cell retrieval efficiency for the RT-qPCR? If the cells were strongly attached on the membrane, it would not be easy to retrieve all cells and the harsh reagents may change the phenotype of the cells. The selection of gene markers should be justified.

Answer #8

Thank you for this comment. The cell seeded 6-well-plates are immediately placed on ice, before harvesting them with QIAZOL. At 4°C the metabolism of the cells and the activity of digestive enzymes is reduced to a minimum. Furthermore, QIAZOL contains Phenol/Guanidin thiocyanate, which leads to cell lysis as soon as it encounters with cells. The phenotype is thus not influenced. Moreover, the harvesting procedure is equal for cells/conditions. Therefore, if there was influence on the cells, it would be same for all conditions.

The selection of genes and molecules was supplemented in an additional section in materials and methods, lines 154-195.

Comment #9

Western blot analysis: For the primary antibodies, were they mono- or polyclonal? The selection of intracellular protein markers should be justified. 

Answer #9 

The selection of genes and molecules was supplemented in an additional section in materials and methods, lines 154-195.

Regarding the primary antibodies, the following changes were made in materials and methods, lines 285-288: After washing, the blots were incubated with the appropriate secondary antibody Goat Anti-mouse IgG H&L (Abcam, Cambridge, UK, abb6789, monoclonal, 1:5000 for α-SMA) and Goat Anti-rabbit IgG H&L (Abcam, abb6721, polyclonal, 1:5000 for COL1A1 and GAPDH).

Comment #10

Luminex: Please provide more details about the procedures and the protocol, e.g., standards, blank controls etc. The selection of the protein markers should be justified. Usually when multiplex assay was run, a validation step of selected markers is needed. Please explain if any validation was done with the Luminex results. If not, why not? 

Answer #10

Thank you for your valuable remark. The selection of genes and molecules was supplemented in an additional section in materials and methods, lines 154-195.

The following modifications were made based on comment #10 in manuscript, materials and methods, lines 295-306: Human Magnetic Luminex® Assays (LXSAHM, R&D Systems) were used for the quantification of selected proteins in the supernatant. The custom-designed multiplex assays measuring COL1A1, FN1, TIMP1, proangiogenic factors (VEGF A, VEGF C), bFGF and IL11 were performed according to the manufacturer’s protocol. Standards for each analyte were provided by the manufacturer and included in the kit. SM without cytokines and SM with cytokines were used as blanks. The sample dilutions for the selected markers were estimated from previous Luminex measurements (e.g. Gugatschka et al 2019, Grossmann et al. 2020), therefore no additional validation experiments were performed. For COL1A1, FN1 and TIMP1 measurement, samples were pre-diluted 1:50; for all other analytes, samples were used undiluted. The assays were measured on the Bio-Plex 200 assay reader (Bio-Rad). Calculations were carried out using the Bio-Plex Manager Software Version 6.2 (Bio-Rad).

Comment #11

ELISA: Please provide more details about the procedures and the protocol, e.g., blank controls, catalogue number etc. The selection of the single HA marker should be justified.

Answer #11

The following modifications were made based on comment #11 in manuscript, materials and methods, lines 166-169 and 309-321: Hyaluronic acid is a ubiquitous glycosaminoglycan and is of particular importance for tissue. It is synthesized via hyaluronan synthases (HAS) of which three isoenzymes HAS1, HAS2 and HAS3 with different kinetic properties are known. Hyaluronidase (HYAL) 2 is responsible for HA degradation.

Hyaluronic acid (HA) levels in the supernatant were determined using the sandwich enzyme immunoassay Quantikine® ELISA Kit (DHYAL0, R&D Systems) according to the manufacturer’s protocol. Briefly, an HA standard provided in the kit was used and diluted with calibrator diluent RT5-18. This diluent also served as the zero standard. An HA standard curve was prepared with HA concentrations from 0.625 to 40 ng/mL. SM without cytokines and SM with cytokines were used as blanks. All samples and blanks were pre-diluted 1:100 with Calibrator Diluent RD5-18 and assayed with HA standards on the same plate in technical duplicates. Optical absorbance (OA) values were measured using the Spectramax Plus 384 Microplate Reader (Molecular Devices), OA values at 540nm were substracted from OA values at 450nm. Sample concentrations were determined from calibration curve, the corresponding blank value was subtracted and resulting numbers were multiplied with the dilution factor (x100). 

Comment #12

Statistics: Please make it clear which variables were treated as "between-subject" independent variables and which comparisons were made in the design.

Answer #12

The following modifications were made based on comment #12 in manuscript, materials and methods, lines 325-327: The four different conditions, static without cytokines, static with cytokines, vibration without cytokines and vibration with cytokines were treated as independent variables.

Comment #13

Results: Please report all results include those of LDH and the TIMP1. Also, the presentation of the results were mixed with PCR, mRNA, Luminex and ELISA. As the choice of markers lacks justification in the methodology, it made the results herein very difficult to follow and comprehend. A reorganization of this section in terms of writing and figures (Figure 2-5) is strongly needed. Please include asterisk for significant values. For the decimal numbers above the bars (Figure 2-5), what did these numbers represent? Were they p-value from ANOVA or post-hoc pairwise comparison? 

Answer #13

The reviewer is right, the results were not arranged by analyzing methods but by classes of molecules. From our point of view, the arrangement of each class of molecule in one paragraph as well as figure is easier and more comprehensible for the reader to follow. We think that, from a didactical point of view, it is more reasonable to stay with this structure. 

Regarding significant values, we refrained from including asterisks in the figures. However, we agree with the reviewer, that the explanation might be misleading, therefore we included the following sentence to the figure legend: Decimal numbers above the bars represent the statistically significant p-values from post-hoc comparisons. For the not presented p-values see the supplementary appendix.

LDH and TIMP1 results were added to the manuscript.

The following modifications were made based on comment #13 in manuscript, results, lines 337-338: No significant differences in LDH-activity were found between the different conditions (p = 0.966).

The following modifications were made based on comment #13 in manuscript, results, lines 379-380: Neither condition influenced TIMP1 expression (p= 0.629; �2= 0.130).

Comment #14

Results: Figure 3. For Col1A1, the pattern of changes for "+ cytokine" between static and dynamic conditions are different for (d) and (e). Why so? Such results should be in sort of agreement.

Answer #14

Unfortunately, we cannot fully comprehend this comment, since no difference in the sort of agreement can be seen in our image. Could the reviewer please specify this in more detail. 

Comment #15

Discussion: In general, please denote clearly which figure that the results were referred to in the main text. Also, some claims were not supported by the results. For example, Line 321-324 claimed that their results showed up-regulated ECM- and inflammation-related genes and proteins. However, Figure 5 showed the downregulation but not the upregulation for ACTA2, TGFbeta 1, IL-11 and alpha-SMA under the "+ cytokine/ dynamic" condition, compared to the "+cytokine/ static" condition.

Answer #15

The following modifications were made based on comment #15 in manuscript, discussion, lines 488-491: In line with these results, the upregulation of ECM- and inflammation-related genes and proteins in our experiments, such as collagen, HA as well as IL1�, TGF-�1 and IL6 reflect the applicability of our model, even though only one cell type was included.

Comment #16

Please include a more thorough discussion around the biological/ physiological functions of IL-11 on fibroblasts and the synergistic interactions of IL-11 with other cytokines.

Answer #16

The following modifications were made based on comment #16 in manuscript, materials and methods, lines 192-195: IL11, another member of the IL6 family, is known to be mechanoresponsive and is also involved in tissue fibrogenesis. It has been shown that this cytokine is a downstream effector of TGFβ1 and contributes to the differentiation of pulmonary fibroblasts to ACTA2 positive myofibroblasts.

Comment #17

Please discuss what the limitations and future direction of this study.

Answer #17

The following modifications were made based on comment #17 in manuscript, discussion, lines 529-536: There are some limitations in the present study. Firstly, even though the bioreactor is able to expose the cells to a wide range of frequencies and to activate some mechanotransductive pathways, we are aware that the mechanical forces during phonation are much more complex and not reproduced to their full extent here. Secondly, only one cell type was used, whereas inflammatory reactions in vivo involve multiple cell types (e.g. macrophages and neutrophil granulocytes). And thirdly, the fact that the VFF are immortalized might lead to different behavior compared to near primary cells or to in vivo cells.

The following modifications were made based on comment #17 in manuscript, conclusion, lines 569-571: Further research should focus on the approximation of in vivo inflammatory conditions by for example co-cultivating VFF with other cell types and developing appropriate three-dimensional models.

Reviewer #2 

The main finding in this paper is that vibration of hVFFs results in reduced expression of pro-inflammatory cytokines and smooth muscle actin while in an induced inflammatory state.The ECM component HA which is necessary for reduction of fibrosis is not influenced by vibration.

The introduction is brief and fails to acknowledge clinical studies addressing vocal rest/use post-surgery, findings which provide necessary context to this manuscript. The methodology is well designed and reflects an appropriate number of replicates. The results are of interest to the field and align with early ambulatory recovery guidelines in other disciplines. The discussion section fails to address any limitations of the study and is oversimplified such that some statements are partially incorrect. If the findings of this study translate to human voice they may have a significant impact on post-surgery guidance for voice use following phonosurgery.

The manuscript would benefit from external copy-editing as there are many instances where the meaning is unclear, or wording is inappropriate. No supplementary materials were available in the reviewer’s download so it is unclear if all the data is freely available.

Comment #1

Title gives the impression this is a review or conducted in humans, should reflect study more accurately.

Answer #1

The title was modified to: Exploring the Pathophysiology of Vocal Fold Inflammation: The Molecular Impact of Vibration.

Comment #2

Abstract: Line 18: word choice - Reword to reflect that biopsies for molecular biology studies are unethical, as biopsies are possible. 

Line 19: word choice - As the model had been established prior (cell line Thiebault, biomechanical reactor in a previous publication) reword to reflect utilization of a previously developed in vitro phonomimetic biomechanical model.

Line 24: word choice - Wording ambiguous as to whether the mechanical stimulus or cytokine stimulus is applied for 4 hours.

Answer #2

The wording was changed as suggested by the reviewer.

Comment #3

This section fails to address the role of voice therapy in the treatment of vocal fold lesions – for most lesions voice therapy is recommended in combination with surgery, and in some cases alone. Omission of voice therapy weakens this section substantially.

Answer #3

Since the changes the introduction were considerable, we refrain from inserting all amendments to this reply. However, all changes are labelled in the text in red and all proposals of the reviewer addressed.

Comment #4

Line 41: word choice - sentence meaning is unclear. Reword to reflect damage caused surgically, or mechanically by phonotrauma lead to an upregulation of inflammatory cytokines.

Line 44: word choice - use of “the worst-case” unnecessary and changes sentence meaning.

Line 46/47: word choice – reword to reflect that these findings may influence clinical practice guidelines.

Answer #4

The wording was changed as suggested by the reviewer.

Comment #5

Line 57: The statement, “barely any sound evidence,” is inappropriate. There are several peer reviewed articles addressing voice rest/use following phonomicrosurgery. See Bercirovic et al. 2019, Whitling et al. 2018, Kiagadaki et al. 2014, and Wang and Huang 1994. The findings of these studies lack consensus and serve to strengthen the authors arguments.

Answer #5

The following modifications were made based on comment #5 in manuscript, introduction, lines 63-70: The recommendation on post-operative voice rest is based on the hypothesis that preserving the tissue improves wound healing. A distinction is made between absolute and relative voice rest. The former entails a complete avoidance of any mechanical stimulation of the VF tissue through phonation. The latter still lacks a clear definition. But as in other medical fields, where a rapid mobilization after surgical interventions is now often recommended to achieve an early recovery, there has been a tendency over the past years, to decrease the interval of voice rest following phonomicrosurgery.

The following modifications were made based on comment #5 in manuscript, introduction, lines 74-80: Furthermore, a shorter period of voice rest means that voice therapy can be started sooner. 

Recent randomized controlled studies could not confirm any beneficial effects of absolute voice rest relative to a reduced postoperative voice use, as measured in terms of subjective parameters (e.g. VHI) as well as acoustic and aerodynamic parameters. There are still different concepts dealing with this issue, and a commonly accepted consensus is not available.

Comment #6

Line 59: word choice – the healing process is influenced/impacted by these factors, “trigger” is not correct in this usage.

Line 65: word choice – consider necessary or essential as opposed to mandatory, changes sentence meaning.

Answer #6

The wording was changed as suggested by the reviewer.

Comment #7

Line 66/67: ECM production changes are inadequately referenced.

Answer #7

We agree with the reviewer. The following modifications were made based on comment #7 in manuscript, introduction, lines 105-106: However, to our knowledge, the impact of vibration on hVFF during inflammation is still unclear.

Comment #8

Line 68: While there is not much evidence, Zhang et al. 2015 describe the impact of cyclic tensile strength on hVFFs in the presence of cigarette smoke and should be included here.

Answer #8

The following modifications were made based on comment #8 in manuscript, introduction, lines 104-105: Zhang et al. demonstrated that the application of cyclic tensile strain on hVFF attenuates the inflammatory reaction induced by cigarette smoke.

Comment #9

Comment #9

9) Line 85: word choice – remove “Therefore.”

Line 94: word choice – sentence does not make sense. Reword to reflect that inclusion of inert macromolecules enhances ECM production such that it is easier for researchers to evaluate them in culture. 

Line 99: word choice – “foreseen” used incorrectly, consider stratified/randomized/ or assigned to throughout the manuscript

Line 119: word choice – reads awkwardly. Consider rewording “orientating on the publication of…” to, “as described by.”

Line 119: double period at end of sentence. 

Line 120: word choice – “foreseen” incorrectly used

Line 152: double period at end of the sentence. Siehe oben!

Answer #9

The wording as well as the punctuations were changed as suggested by the reviewer. 

Comment #10

Line 139/140: How was the RNA quality established? No mention of RIN or the use of a denaturing agarose gel. This data is essential to ensuring the RNA meets quality standards for qPCR. 

Answer #10

Thank you for this valuable comment. By our experiences from previous experiments, the RNA quality from cell cultures is very high. The determination oft he RIN is of utmost importance in tissue analysis, such as biopsies. Since we used immortalized vocal fold fibroblasts we refrained from a RNA quality analysis. However the purity was calculated and added.

The following modifications were made based on comment #10 in manuscript, materials and methods, lines 246-248: RNA purity was established by calculating the ratio of the absorbance at 260nm and 280nm. A260/A280 > 1,8 was considered to be clean.

Comment #11

Line 144: Please include more details for the reaction (number of cycles, temperature, nucleic acid dye, etc.) Looking at the paper referenced in this section this too lacks details about the qPCR methods.

Answer #11

The following modifications were made based on comment #11 in manuscript, materials and methods, lines 250-256: 10ng in 4µL of cDNA were mixed with 6µL of Primer-SYBR Master Mix (Promega/USA) consisting of 1µL (200nM each) Primer F&R and 5µL GoTaq Master Mix (Promega/USA). RT-qPCR was performed in technical triplicates using the LightCycler 480 system (Roche, Vienna, AT). The following program was applied for denaturation, amplification and the melting curve analysis, respectively: 2 min/95 °C, 45 cycles of 10 s/95 °C and 1 min/60 °C, followed by constantly increasing the temperature at a rate of 2.5 °C/min from 55 °C to 95 °C.

Comment #12

Include effect sizes in the manuscript as these may demonstrate clinical implications where non-significant “trends” are observed. Consider Gaeta and Brydges 2020 JSLHR article when reporting effect size. 

Answer #12

Effect sizes were included in the entire results (changes are marked in red). Gaeta and Brydges refer to Cohen’s d and Hedges g and then calculate 25th, 50th and 75th percentiles. Since we conducted a 1-way ANOVA, we calculated partial eta squared (partial �2) and therefore did not discuss the mentioned publication.

Comment #13

General: Include p values and effect sizes in the text describing the results. We included both of them in the Figure legends but not the post hoc values (these are included in the figures) in order to not to disturb the flow of reading.

General: Include p values and effect sizes in the Figure legends.

Answer #13

The p-values from the 1-way ANOVA as well as the effect sizes were supplemented in the manuscript. However, for a better overview, the p-values from the post hoc tests were only described in the figure legends.

Comment #14

General: This section would benefit from external copy-editing. A number of sentences are unclear in their meaning, and there is a repeated use of some phrases which read unclearly or strangely, e.g. failed statistical significance, not altered by.

Answer #14

An external copy editing was performed.

Comment #15

Line 213: word choice – change “under” to with.

Line 214: word choice – consider rewording to, “Gene expression results were consistent with significantly elevated transcript levels of HAS1 and HAS3.”

Line 231: word choice – reword “failed statistical significance” to something like “did not meet statistical significance” throughout.

Line 246: word choice – insert, “and was” before, “not altered by additional…”

Line 265 – 267: word choice – sentence meaning is unclear. Do the authors mean that non-significant upregulation was observed with cytokine treatment alone?

Line 274: word choice – reword, “under” to, “in the.”

Answer #15

The wording was changed as suggested by the reviewer. 

Comment #16

Line 262: Figure legend for Fig 4 is incomplete.

Answer #16

The Figure legend was corrected for better understanding as follows: mRNA (a) and protein levels (b-d) of ECM-related molecules as well as growth- and angiogenic factors were analyzed by RT-qPCR and Luminex, respectively.

Comment #17

The use of SYBR for qPCR. SYBR dye binds to double stranded DNA in a non-specific manner, thus does not confirm that the correct transcript has been replicated. Use of a Dual-labeled Primer (e.g. TaqMan) is more specific and reproducible.

Answer #17

Thank you for this important suggestion. We agree, that SYBRgreen dye binds nonspecifically to double stranded DNA. Therefore, there is a certain risk that genomic DNA and undesired sequences distort the results. In order to avoid these issues, we add DNAse for removing genomic DNA contamination from RNA samples prior to reverse transcription. Furthermore, we perform melting curve analysis to identify amplifications of undesired DNA sequences.

Comment #18

Immortalized cell lines lack a bloody supply, and thus inflammatory observations are made outside of a normal physiologic system and may not be translatable to in vivo.

Immortalized cell lines have inherent property changes from primary cells or in vivo cells and may behave differently.

In vitro biomechanical manipulation is an approximation of voice and may not apply identical shearing and stress forces as observed in physiologic voice

Answer #18 

Thank you for this very important comment. 

The following modifications were made based on comment #18 in manuscript, discussion, lines 529-536: There are some limitations in the present study. Firstly, even though the bioreactor is able to expose the cells to a wide range of frequencies and to activate some mechanotransductive pathways, we are aware that the mechanical forces during phonation are much more complex and not reproduced to their full extent here. Secondly, only one cell type was used, whereas inflammatory reactions in vivo involve multiple cell types (e.g. macrophages and neutrophil granulocytes). And thirdly, the fact that the VFF are immortalized might lead to different behavior compared to near primary cells or to in vivo cells.

Comment #19

Line 292: The role of inflammation is grossly understated. Inflammation is also required for homeostasis/adaptation to stress/promotion of healing/recruitment of host immune response, etc.

Answer #19

Thank you for this suggestion. 

The following modifications were made based on comment #19 in manuscript, discussion, lines 449-451: The early phase of tissue repair is accompanied by an inflammatory reaction, which among other things promotes wound healing, recruits inflammatory cells and can preempt an infection in the region of injury.

Comment #20

Line 319: Italicize in vivo.

Line 340: spelling incorrect – glycosaminoglycan

Line 346: word choice – use of, “could” is ambiguous, consider, “demonstrated an increase…”

Line 352: word choice – use of, “could” is ambiguous, consider, “did not…”

Line 354: Please change ex vivo to in vitro. Ex vivo implies minimal changes to the tissue upon removing it from the body.

Line 387: italicize in vivo

Answer #20

The wording, spelling as well as the formating was changed as suggested by the reviewer. 

Comment #21

Line 326: Statement is incorrect. Rousseau model is a controlled vibratory phonation model using electrical stimulation to elicit vocalization. See Kimball et al. 2019 for information on stimulation and vibratory patterns in this model.

Answer #21

Thank you for this comment. The mentioned statement was deleted from the manuscript.

Comment #22

Figures: Consider removing the p values from the figures and use asterisks to denote the level of significance. The p values make the graphs seem over complicated. 

Answer #22

Thank you for your suggestion. We refrained from including asterisks in the figures, since we think this facilitates the overview. However, for a better understanding, we included the following sentence in the figure legends: Decimal numbers above the bars represent the statistically significant p-values from post-hoc comparisons. For the not presented p-values see the supplementary appendix.

Reviewer #3

This is an interesting paper describing application of a sophisticated bioreactor for studying inflammatory signaling in VFFs in concert with a vibratory stimulus. The bioreactor technology has previously been published by the group. This work is certainly a useful contribution to the literature, and I offer the following comments and suggestions:

Comment #1

Is the vibratory dose delivered to the 6 well plate uniform across all wells, or does it vary based on the position of the well? If there is well-to-well variation, how significant is this and how was it managed?

Answer #1

We are very thankful for this comment. We cannot fully exclude slight variations, which are due to productional deficiencies of the six-well-plates. However, in order to avoid impacts on the results, we pooled the harvested materials. This method has proven itself very successful in our previous publications: 

- Kirsch A, Hortobagyi D, Stachl T, Karbiener M, Grossmann T, Gerstenberger C, et al. Development and validation of a novel phonomimetic bioreactor. PLoS ONE. 2019

- Grossmann T, Steffan B, Kirsch A, Grill M, Gerstenberger C, Gugatschka M. Exploring the Pathophysiology of Reinke’s Edema: The Cellular Impact of Cigarette Smoke and Vibration. Laryngoscope. 2020 Jun 22

Comment #2

MMC was used in all conditions to enhance biological responsiveness and more closely simulate a fibrosis situation, however this is just mentioned briefly in the Method section. It would be very helpful to show the effect of MMC on the cytokine and vibration responses here – ideally with a non-MMC experimental condition or, if not practical, by describing the results in light of this groups’ prior publication that showed the influence of MMC alone on VFF biology in vitro. This really should have more emphasis, especially when the authors interpret and compare their results to those of others who did not use MMC in their experiments.

Answer #2

Thank you for this important suggestion. The following modifications were made based on comment #2 in manuscript, discussion, lines 537-545:

Another problem of in vitro experiments is the aqueous environment surrounding the cells. In vivo, cells are embedded in a highly crowded environment with plenty of macromolecules known to have equally important kinetic and thermodynamic functions such as setting the optimal pH. Graupp et al. demonstrated in a VFF model that macromolecular crowding in combination with TGF-� achieves a significantly higher concentration of ECM components (e.g. collagen and HA) as well as of �-SMA compared to an uncrowded control. For this reason the experiments presented here were performed under crowded conditions. A direct comparison with other studies which disregard macromolecular crowding is not possible.

Comment #3

I appreciate the care given to Tukey adjustment and parametric versus non-parametric statistical test selection, but wonder if these data would be best handled by a 2-way ANOVA in which cytokine treatment and vibration treatment are handled as independent variables, with their interaction effect included. The interaction between these experimental stimuli may be important here. Also, with no word or page limits in this online journal, I please report the results of the F and KW tests (not just the significant pairwise comparisons), as well as the nonsignificant LDH data. These can easily be placed in a supplementary materials section. 

Answer #3

We totally agree that the interaction of cytokines and vibration is important and a 2-way ANOVA would also be absolutely valid. But to a certain degree it is subject to interpretation. In this study we particularly wanted to evaluate the difference between the two conditions with cytokine treatment (static vs. dynamic). Since there are only two levels in each variable (with vs. without cytokines - static vs. dynamic), a post hoc comparison in 2-way ANOVA was not possible. Therefore, we decided to treat each of the four conditions as independent variables and applied a 1-way ANOVA. 

The results of the F and KW tests as well as the results from the LDH-assay were added in the supplementary appendix and the manuscript, respectively.

The following modifications were made based on comment #3 in manuscript, results, lines 337-338: No significant differences in LDH-activity were found between the different conditions (p = 0.966).

Comment #4a

Finally, I suggest more careful and conservative description of the clinical and wound healing issues that motivate the work and are described in the introduction and discussion. For example:

Early inflammation is not just for avoiding infection (start of discussion). There is much more complex biology involved. 

Answer #4a

The following modifications were made based on comment #4a in manuscript, discussion, lines 449-451: The early phase of tissue repair is accompanied by an inflammatory reaction, which among other things promotes wound healing, recruits inflammatory cells and can preempt an infection in the region of injury.

Comment #4b

I disagree that there is a lack of clinical recommendations for voice rest (Abstract); rather there is a lack of UNIFORM recommendations.

Answer #4b

The following modifications were made based on comment #4b in manuscript, abstract, lines 16-18: 

Voice rest following phonotrauma or phonosurgery has a considerable clinical impact, but clinical recommendations are inconsistent due to inconclusive data.

Comment #4c

The comparison of VF vibration with orthopedic mobilization after injury or surgery is a more complex and nuanced comparison than is suggested (Introduction). These are quite different mechanical situations and I suggest describing in more detail.

Answer #4c

Thank you for this important suggestion. The following modifications were made based on comment #4c in manuscript, introduction, lines 70-74: Obviously, the objectives of rehabilitation of, for example, a joint and the VF are not identical. While resilience is more important in orthopedics, pliability is the predominant aim in VF wound healing. Nevertheless, early mobilization is known to have antifibrotic effects, which might also be useful in the field of laryngology.

Comment #4d

I suggest citations to support “size of the VF lesion, smoking status, sex, age etc” (Introduction). 

Answer #4d

The mentioned sentence was deleted in the introduction, due to multiple changes proposed by other reviewers. 

Comment #4e

Paragraph 2 in the Discussion might fit better in the Introduction. 

Answer #4e

Due to various revision proposals of other reviewers, we think that this paragraph is more conclusive in the discussion.

Comment #4f

I think that “in vivo” (line 387, Conclusion) was intended to be “in vitro”.

Answer #4f

Thank you calling our attention to this important point. The following modifications were made based on comment #4f in manuscript, conclusion, lines 567-569: Notwithstanding the inherent limitations of an in vitro study, it might suggest that a certain amount of postoperative vocal fold vibration might have a beneficial impact on wound healing.

Reviewer #4: The purpose of the current study was to explore the effects of vibration on hVFF in an inflammatory and normal state. I very much like the premise of this study as it provides an important avenue to discuss biological outcomes associated with mechanical stimulation following inflammation. However, I have concerns regarding manuscript overall clarity, methodological decisions, and the conclusion. Specific comments are listed by section.

Comment #1

I would consider changing the title of the manuscript. First, it sounds like the paper is a Review as opposed to an experimental study. Also, it is truly not reflective of what was done in the study. Vocal folds were not injured. VFF were treated with an inflammatory challenge.

Answer #1

The title was modified to: Exploring the Pathophysiology of Vocal Fold Inflammation: The Molecular Impact of Vibration.

Comment #2

The Introduction statement of the abstract is not clear. The important role of voice rest is introduced, but then stated that the authors seek to evaluate the effect of vibration on VFF. 

Answer #2

Thank you for your comment. A plethora of changes throughout the manuscript were performed to further elucidate this point.

Comment #3

Not all readers will be familiar with voice rest recommendations following phonosurgery. The Introduction would benefit from a brief overview (1-2 sentences) of typical voice rest recommendations. This would provide some context for the reader.

Answer#3

Thank you for this important suggestion. The following modifications were made based on comment #3 in manuscript, introduction, lines 63-67: The recommendation on post-operative voice rest is based on the hypothesis that preserving the tissue improves wound healing. A distinction is made between absolute and relative voice rest. The former entails a complete avoidance of any mechanical stimulation of the VF tissue through phonation. The latter still lacks a clear definition.

Comment #4

The authors begin the Introduction discussing upregulation of inflammatory cytokines following phonotrauma, but not mention of the inflammatory cascade following phonosurgery. This should be mentioned as I think the authors are making of case to study the role of voice rest following phonotrauma and/or vocal fold surgery.

Answer #4

The following modifications were made based on comment #4 in manuscript, introduction, lines 42-47: Dermal studies have shown that interleukin (IL) 1 is the body’s first signal of tissue damage. This leads to a degranulation of thrombocytes and consequently to a release of different growth factors such as transforming growth factor β (TGF β), platelet-derived growth factor (PDGF) and epidermal growth factor (EGF). These factors in turn activate a plethora of other cell types, thereby inducing inflammation and wound healing.

Comment #5

The Introduction needs further information (1-2 sentences) regarding the important role of VFF in vocal fold physiology.

Answer #5

The following modifications were made based on comment #5 in manuscript, introduction, lines 101-103: The VFF is predominantly responsible for the composition of the lamina propria and is therefore essential for smooth phonation.

Comment #6

In general, it is unclear that the authors are trying to mimic inflammation following phonotrauma or phonosurgery and the subsequent role of vibration versus rest.

Answer #6

The following modifications were made based on comment #6 in manuscript, introduction, lines 112-115: By combining these factors, we hope to mimic inflammation after phonosurgery as accurately as possible, in order to better understand the effects of vibration versus (voice) rest on ECM-related molecules and angiogenic factors as well as on inflammatory and fibrogenic markers.

Comment #7

The objective of the study is listed as studying the cellular responses of hVFF following a profibrotic and inflammatory stimulus under static and dynamic conditions. I have several concerns with this statement. First, cellular responses are extremely broad and can encompass a huge range of factors. I would recommend expanding upon this. Furthermore, the authors state in the title that they are seeking to study the molecular, not cellular perspective. It is unclear that static and dynamic is supposed to related to vocal fold rest and vibration, respectively. Finally, while inflammation if briefly introduced, what is the importance of a pro-fibrotic stimuli? 

Answer #7

Thank you for this valuable comment. We are aware of the fact that cellular responses are diverse, however we focused on ECM related proteins, proangiogenic factors and proinflammatory as well as profibrotic markers.

In our opinion, molecular and cellular changes come along with each other and cannot be viewed separately.

Regarding static and dynamic conditions the following sentence was added for a better understanding, as mentioned for answer #6.

While an inflammation is temporary, its consequences lead to tissue damage, such as fibrosis and, in case of the VF, to a deterioration of the oscillatory properties. This has been discussed in the introduction (lines 48-53).

Comment #8

Please provide justification regarding the choice to serum starve the VFF. Serum starvation has been shown to cause cells to undergo apoptosis and increase cell susceptibility to inflammatory stimuli. 

Answer #8

The following modifications were made based on comment #8 in manuscript, discussion, lines 466-470: As mentioned earlier, VFF were exposed to a serum free medium for 24 hours. It is known that starvation influences the susceptibility of cells to inflammatory stimuli. On the other hand, it is a well-established method for making cell populations more homogeneous and improving reproducibility. This method has already been applied in previous vocal fold fibrosis models.

Comment #9

The term pro-fibrotic and inflammatory culture conditions is utilized. While the use of the inflammatory makes sense, pro-fibrotic is confusing. Please provide justification / citations regarding the pro-fibrotic effect of these cytokines. 

Answer #9 

There are many studies which confirm the profibrotic effect of TGFb. Some of them are listed below:

-Transforming growth factor beta (TGF-b) isoforms in wound healing and fibrosis 

Michael K. Lichtman, MD; Marta Otero-Vinas, PhD; Vincent Falanga, MD, FACP (cited in the manuscript)

-Transforming growth factor β1 (TGF-β1) enhances expression of profibrotic genes through a novel signaling cascade and microRNAs in renal mesangial cells

Nancy E Castro, Mitsuo Kato, Jung Tak Park, Rama Natarajan

-TGF-β-induced profibrotic signaling is regulated in part by the WNT receptor Frizzled-8

Anita I R Spanjer, Hoeke A Baarsma, Lisette M Oostenbrink , Sepp R Jansen, Christine C Kuipers, Michael Lindner, Dirkje S Postma, Herman Meurs, Irene H Heijink, Reinoud Gosens, Melanie Königshoff 

Comment #10

It is stated that these cytokines were capable of maintaining a consistent inflammatory reaction for at least 72 hours. How was this measured / verified?

Answer #10

Prior to the present study we conducted trials to evaluate whether TGFβ1, IL1β or a combination of both was the most appropriate to simulate an acute inflammatory reaction. This was described in more detail in the added supplementary appendix.

Comment #11

I realize that the cells were only vibrated for four hours, but in the initial section regarding cell culture and treatment it reads like the cells were vibrated for 72 hours.

Answer #11

Thank you for this advice. 

The following modifications were made based on comment #11 in manuscript, materials and methods, lines 145-146: Vibration was applied for four hours daily as described below.

Comment #12

In Mechanical Stimulation, I know that a citation is provided, but it would still benefit from 1-2 sentences justifying how vibration parameters and time relate to human voice production. 

Answer #12

The following modifications were made based on comment #12 in manuscript, materials and methods, lines 210-220:The stimulatory pattern, a linear sinusoidal chirp sound in a frequency range of 50 Hz to 250 Hz corresponds to the frequency range of the primary larynx sound in humans. The membrane displacement in the center of a well was dependent on the applied frequency and averaged 82µm ± 5µm. The input voltage (V) on the loudspeaker was set to 1.1 V and served as a measure of the vibration intensity. Stimulation was applied for four hours per day, following the publication of Titze et al. There are a handful of publications dealing with displacements (vertical and horizontal) of human vocal folds during phonation, reporting a wide range of values (depending on the setting and/or measurement method). The simulation of these displacements is reproducible with our bioreactor and complies with the maximum displacement in humans. In order to rule out any interferences, cells assigned to the static group were kept in a separate incubator.

Comment #13

It is unclear to me how the cell viability assay was conducted. The authors state to evaluate maximum LDH activity, cells were seeded in parallel….. Were the cells seeded after mechanical stimulation for the purpose of this assay? How would that potentially affect the viability results? Please clarify.

Answer #13

The section LDH assay was expanded in order to get a more detailed understanding of the working steps performed. Since the changes to the LDH assay were considerable, we refrain from inserting all amendments to this reply. However, all changes are labelled in the text in red.

Comment #14

In addition, the authors state that LDH activity of the samples was expressed as percentage of the maximal LDH activity. However, this data is never presented.

Answer #14

Thank you for this comment. The data and the figure were added to the manuscript.

The following modifications were made based on comment #14, #16 and #22 in manuscript, results, lines 339-340: No significant differences in LDH-activity were found between the different conditions (p = 0.966).

Comment #15

There is no justification in the Introduction or Methods regarding the choice to evaluate various classes of genes and proteins. It is not until the Discussion that the importance of looking at these factors is even mentioned, albeit relatively briefly. This should really come earlier in the manuscript and be expanded.

Answer #15

Thank you for this very valuable suggestion. The following modifications were made based on comment #15 in manuscript, materials and methods, lines 154-195:

Selection of genes and molecules

ECM-related molecules

ECM-related molecules describe components of the extracellular scaffold and enzymes that are responsible for the ECM homeostasis and therefore essential in scar development. The two most prevalent ECM components of the VF lamina propria are collagen and HA.

Procollagen, the progenitor molecule, is secreted from the cell and undergoes modifications that lead to the formation of a crosslinked network of fibrils. The most abundant types of collagen in the VF lamina propria are type 1 and type 3. Matrix metalloproteinase (MMP) 1 is responsible for the degradation of collagen type 1 and 3. Tissue inhibitor of metalloproteinase (TIMP) on the contrary is considered to be the antagonist of MMP. 

Hyaluronic acid is a ubiquitous glycosaminoglycan and is of particular importance for tissue. It is synthesized via hyaluronan synthases (HAS) of which three isoenzymes HAS1, HAS2 and HAS3 with different kinetic properties are known. Hyaluronidase (HYAL) 2 is responsible for HA degradation. 

Additionally, fibronectin (FN) was included in the present investigations because it was known to play an important role in cell-to-cell and matrix-to-cell adhesion. 

Proangiogenic factors

A previous investigation of the effect of cigarette smoke extract on VFF by our group showed a significant upregulation of vascular endothelial growth factors (VEGF) A and C. In order to evaluate potential changes on VEGF A and C following cytokine treatment and vibration, these two factors were included in the present study. 

Inflammatory and fibrogenic markers

In this study it was only possible to investigate a limited subset of cytokines, in the knowledge that many more could be relevant.

Alpha smooth muscle actin (ACTA) 2 is the encoding gene for the myofibroblast marker α-SMA. This protein contributes to vocal fold tissue contraction and thus to vocal fold fibrogenesis. 

IL1β, IL6, cyclooxygenase (COX) 2 and TGFβ1 were selected since various studies have previously demonstrated their decisive role in acute inflammatory reactions. 

In in vitro experiments, basic fibroblast growth factor (bFGF) had an anti-inflammatory effect by downregulating collagen production and upregulating HA production. 

IL11, another member of the IL6 family, is known to be mechanoresponsive and is also involved in tissue fibrogenesis. It has been shown that this cytokine is a downstream effector of TGFβ1 and contributes to the differentiation of pulmonary fibroblasts to ACTA2 positive myofibroblasts.

Comment #16

Although NS, I feel the manuscript would benefit from inclusion of the viability data.

Answer #16

The data and the figure were added to the manuscript.

The following modifications were made based on comment #14, #16 and #22 in manuscript, results, lines 339-340: No significant differences in LDH-activity were found between the different conditions (p = 0.966).

Comment #17

Subheading including ECM, angiogenic factors, fibrogenic markers are first introduced in the Results section. These are important classes of genes and proteins and it is strange to see these subheadings for the first time in the Results section. 

Answer #17

The classes and their justifications were included in materials and methods section (lines 154-195) as described for answer #15.

Comment #18

I find the results very difficult to follow. Specifically, the most important findings (e.g. IL11) are difficult to realize, because so many genes are discussed typically individually. I believe there some redundancies were the same genes demonstrate the same findings in terms of significance related to inflammatory condition and vibration. I wonder if there is a way for the authors to condense / re-organize.

Answer #18

Thank you for this important comment. From our point of view, the arrangement of each class of molecule in one paragraph as well as figure is easier and more comprehensible for the reader to follow. We think that, from a didactical point of view, it is more reasonable to stay with this structure. 

Comment #19

The authors frequently use the terms trending towards significance, near significance etc. I would suggest being more judicious with the use of these terms. These are not significant findings. Any interesting trends the authors may wish to save for the Discussion.

Answer #19 

This is an absolutely valuable point. We corrected these terms throughout the manuscript. All changes are labelled in the text in red.

Comment #20

The overall setup for the manuscript is much clearer in the first paragraph of the Discussion than any other part of the manuscript. 

Answer #20

Thank you for this important comment. We have corrected many parts in the manuscript for a clearer structure and a better understanding.

Comment #21

The authors indicate that different inflammatory and pro-fibrotic stimuli were investigated based on previous publications of in vitro inflammation. Were these studies with VFF or other cell types? Furthermore, on page 17 the authors discuss inflammatory cytokines following vocal fold injury in vivo. This Discussion should be moved up and used to justify choice of inflammatory challenge.

Answer #21

Thank you for this suggestion. The supplementary appendix was added for further justification of the inflammatory stimulus. Consequently, the mentioned justification on page 17 was not moved up.

The mentioned studies of Branco et al. and Branski were performed on VFF, the other two on other types of fibroblasts (NIH3T3 fibroblasts and Neonatal human dermal fibroblasts). 

Comment #22

Why is there not mention of the viability data? It is interesting to me that cytokine treatment or vibration did not affect cell viability.

Answer #22

The data and the figure were added to the manuscript. An extensive discussion was not added to the manuscript since an effect on cell viability was not intended here.

The following modifications were made based on comment #14, #16 and #22 in manuscript, results, lines 339-340: No significant differences in LDH-activity were found between the different conditions (p = 0.966).

Comment #22

IL1B was significantly increased in the inflammatory dynamic condition. Why was this finding not discussed? 

Answer #22

This is correct, IL1b was significantly increased in the inflammatory dynamic group compared to the two non-inflammatory groups, however this was intended as justified in the materials and methods (lines 132-139) and discussion (lines 452-458). But it was not significantly increased compared to the static non-inflammatory group.

Comment #23 & #24

The authors state in the current study that no deteriorating effects of vibration were observed on two important ECM components. Were deleterious effects expected? The vibration parameters were not supposed to be traumatic, correct? 

In the conclusion, the authors state “Notwithstanding the inherent limitations of an in vivo study, it might suggest that a certain amount of postoperative/-traumatic vocal load might have beneficial impact on wound healing.” First, this is not an in vivo study. Second, are the authors suggesting that traumatic vocal load might have a beneficial impact on wound healing. There is a significant difference between traumatic vocal load and likely restorative vocal fold vibration. I do not feel comfortable or support the statement that traumatic load is good for wound healing. Was the mechanical stimulation in the current study meant to mimic a traumatic load? This statement needs to be clarified.

Answer #23 & #24

We did not expect any deleterious effects on ECM components, however we wanted to emphasize this point.

The vibration was not meant to be traumatic. Thank you for this comment. 

The following modifications were made based on comment #23 and #24 in manuscript, conclusion, lines 567-569: Notwithstanding the inherent limitations of an in vitro study, it might suggest that a certain amount of postoperative vocal fold vibration might have a beneficial impact on wound healing.

Comment #25

In every Figure caption the statistical analysis methods are stated. This is redundant and in the manuscript text. I suggest removing. 

Answer #25

This is absolutely right, there are some redundancies in the figure descriptions, however we find it is important to mention some points in each graph for quicker and better understanding.

Comment #26

Please proofread carefully. Numerous typos and inconsistencies are appreciated throughout the manuscript.

Answer #26

External copy editing was performed.

---

## [Decision Letter · Decision Letter 1]

23 Sep 2020

PONE-D-20-19358R1

Exploring the Pathophysiology of Vocal Fold Inflammation: The Molecular Impact of Vibration

PLOS ONE

Dear Dr. Hortobagyi,

Thank you for submitting your manuscript to PLOS ONE. After careful consideration, we feel that it has merit but does not fully meet PLOS ONE’s publication criteria as it currently stands. Therefore, we invite you to submit a revised version of the manuscript that addresses the points raised during the review process.

Please respond to all additional comments from reviewers.

We look forward to receiving your revised manuscript.

Kind regards,

Marie Jetté

Academic Editor

PLOS ONE

Reviewers' comments:

Reviewer's Responses to Questions

**Comments to the Author**

1. If the authors have adequately addressed your comments raised in a previous round of review and you feel that this manuscript is now acceptable for publication, you may indicate that here to bypass the “Comments to the Author” section, enter your conflict of interest statement in the “Confidential to Editor” section, and submit your "Accept" recommendation.

Reviewer #1: (No Response)

Reviewer #2: (No Response)

Reviewer #3: (No Response)

2. Is the manuscript technically sound, and do the data support the conclusions?

Reviewer #1: Yes

Reviewer #2: Yes

Reviewer #3: Yes

3. Has the statistical analysis been performed appropriately and rigorously? 

Reviewer #1: Yes

Reviewer #2: Yes

Reviewer #3: Yes

4. Have the authors made all data underlying the findings in their manuscript fully available?

Reviewer #1: Yes

Reviewer #2: Yes

Reviewer #3: Yes

5. Is the manuscript presented in an intelligible fashion and written in standard English?

Reviewer #1: No

Reviewer #2: Yes

Reviewer #3: No

6. Review Comments to the Author

Reviewer #1: The authors addressed most of reviewers' comments. However, a few more suggestions are recommended.

(1) Abstract: Although IL-1beta is considered a typical inflammatory stimulus, TGF-beta is a pro-fibrotic stimulus. Line 24 will need to revise to reflect better the roles of these two cytokines.

(2) Introduction: Some writing edits will be needed. For example. Line 92-99. The transition from introducing Gaston et al and Titze et al' studies and then jumped to Latifi et al.'s bioreactor is abrupt.

(3) Introduction. Line 114-115. It is also awkward to mention angiogenic factors in the conclusive statement without reviewing the roles of angiogenic factors in the earlier paragraphs.

(4) Methods: Line 205. Suggest to remove the word "novel" because the work is already published. If the authors would like to retain the word, they need to justify further what the novelty of their bioreactor was in the paper.

(5) Methods: Line 304. Fix grammatical errors on: "for all other analytes, samples were used undiluted." Also the rationale of not doing validation experiments was weak. As every Luminex plate and experiments may have technical variability, the assumption from previous experiments shall not be applied directly to new experiments. Suggest to run conventional ELISAs for 1-2 markers to confirm the Luminex results.

(6) Results: The presentation/ writing will still need to improve. Often times, when reporting certain genes/ proteins have a significant or insignificant change, the authors did not mention the change in relative to which condition/ group. It causes lots of ambiguity in interpreting the results.

(7) Discussion. Line 555-561. The discussion on the roles of IL11 remains brief and superficial. Also, the subsequent sentence of discussing mechanical stimulation and anti-inflammatory effects seemed to be irrelevant. The logical flow of the argument herein needs to be strengthened further.

(8) Discussion. New studies have revealed the important immunological roles of VF fibroblasts, which should be discussed and cited in the paper. For example:

Foote, Al. G., et al. "Tissue specific human fibroblast differential expression based on RNAsequencing analysis." BMC genomics 20.1 (2019): 308.

King, S. N., et al. "Vocal fold fibroblasts immunoregulate activated macrophage phenotype." Cytokine 61.1 (2013): 228-236.

Li‐Jessen, N. et al. "Cellular source and proinflammatory roles of high‐mobility group box 1 in surgically injured rat vocal folds." The Laryngoscope 127.6 (2017): E193-E200.

Reviewer #2: The manuscript has been thoughtfully revised. Please can you address the following:

1) The new title does not address the concern from the previous review that it was too broad and reads like a review. Please revise this to be study specific. e.g. in vitro mechanical vibration down-regulates pro-inflammatory and pro-fibrotic signalling in human vocal fold fibroblasts.

2) Nomenclature: In vivo should be italicized on line 393. Human genes and transcripts should be italicized (line 183, 353, 354, 355, 393, 431, 484, 485, 503, 511, 514, 548). Reword line 490 as nomenclature differs between gene and protein - if you list the full name of the gene/protein this would be fine.

3) Amend line 200 to "After a further 73 hours..."

Reviewer #3: Overall, the article is much improved. The increased level of detail helps highlight this unique and important contribution. Grammatical errors have been corrected.

I have some minor remaining suggestions:

The organization of the Introduction could be improved. Some very short paragraphs were added about the various models; consequently, the revised Introduction does not have good cohesion and flow. I worry that this section seems unfocused: perhaps organization by subheadings would help.

For example,

Injury/Inflammation

Vocal rest (vs. vocalization)

Bioreactor systems

The statement that therapy “acts primarily as an adjunctive therapy” is not correct with respect to benign lesions [line 55]. Therapy is a primary treatment for nodules and preferred over surgery, for example.

The current level of detail in the Method section is welcome.

What is the meaning of the “two most prevalent ECM components” of the VF LP being collagen and HA [line 158]? If “most prevalent” means “most abundant”, then this statement is incorrect. HA is often touted as being functionally important; however, it represents less than 1% of the LP (per total tissue protein). Refer PMID: 16514800.

The Discussion could use attention to organization and flow also. The authors jump back-and-forth between their findings, some clinical studies, animal studies, a bioreactor study – it is challenging to read and grasp the sequence and logic of the authors’ arguments here. It is especially important to take care because the authors shouldn’t overinterpret their data – which reflect in vitro cell phenotypes under certain conditions (admittedly in a sophisticated experimental set-up) – by implying that they are directly comparable to the results of a clinical vocal rest study, for example. Perhaps subheadings would help here also.

The addition of the paragraph on serum starvation is welcome, however it is jarring and seems out-of-place so early in the Discussion. It seems a more natural fit alongside the limitations of the work, later in the Discussion section.

7. PLOS authors have the option to publish the peer review history of their article (what does this mean?). If published, this will include your full peer review and any attached files.

Reviewer #1: No

Reviewer #2: No

Reviewer #3: No

---

## [Author Response · Author response to Decision Letter 1]

7 Oct 2020

Reviewer #1: 

Comment #1 

Abstract: Although IL-1beta is considered a typical inflammatory stimulus, TGF-beta is a pro-fibrotic stimulus. Line 24 will need to revise to reflect better the roles of these two cytokines.

Answer #1 

The following modification was made based on comment #1 in the manuscript, abstract, lines 25-26: 

Inflammatory and pro-fibrotic stimuli were induced by interleukin (IL)1� and transforming growth factor (TGF)�1, respectively.

Comment #2

Introduction: Some writing edits will be needed. For example. Lines 92-99. The transition from introducing Gaston et al and Titze et al' studies and then jumped to Latifi et al.'s bioreactor is abrupt.

Answer #2

Thank you for this comment. The following modification was made based on comment #2 in the manuscript, introduction, lines 102-111: 

Besides these ‘two-dimensional’ models, there are some three-dimensional approaches as well. Gaston et al. and Titze et al., for example, developed three-dimensional scaffolds where cells are predominantly exposed to tensile stresses. 

Conversely, Latifi et al. developed a bioreactor, which is currently closest to the in vivo situation. Their intention was to imitate all mechanical forces, VFF are exposed to during phonation. They injected a scaffold, consisting of human VF fibroblasts (hVFF), into silicone VF replicas. Cells were exposed to mechanical forces by applying an airflow from beneath which creates a similar mode of vibration to that of the human VF in the larynx.

Comment #3

Introduction. Lines 114-115. It is also awkward to mention angiogenic factors in the conclusive statement without reviewing the roles of angiogenic factors in the earlier paragraphs.

Answer #3

Thank you for this valuable comment. The following paragraph was included based on comment #3 to the manuscript, introduction, lines 50-52:

Since these processes increase the energy metabolism of the tissue, a sufficient blood and consequently nutrient supply is essential. This is enabled by angiogenic factors, which contribute to vessel formation.

Comment #4

Methods: Line 205. Suggest to remove the word "novel" because the work is already published. If the authors would like to retain the word, they need to justify further what the novelty of their bioreactor was in the paper.

Answer #4

The following modification was made based on comment #4 in the manuscript, materials and methods, lines 218-219: 

Mechanical stimulation was performed using a phonomimetic bioreactor developed and established by our group.

Comment #5

Methods: Line 304. Fix grammatical errors on: "for all other analytes, samples were used undiluted." Also the rationale of not doing validation experiments was weak. As every Luminex plate and experiments may have technical variability, the assumption from previous experiments shall not be applied directly to new experiments. Suggest to run conventional ELISAs for 1-2 markers to confirm the Luminex results.

Answer #5

Thank you for this valuable comment. Just to clarify: Of course, we prepared new standard curves for each protein to enable calculation of the concentrations for each assay, according to the manufacturer’s protocol. Only the sample dilution to reach a concentration within the standard curves was estimated from previous studies.

Additionally, we added the standard curves to the supplementary appendix.

Following modification was made based on comment #5 in the manuscript, materials and methods, lines 309-316:

Standards for each analyte were provided by the manufacturer and included in the kit. These standards were used to prepare standard curves (see supplementary appendix). Additionally, SM without cytokines and SM with cytokines was used as blank, respectively. The sample dilutions for the selected markers were estimated from previous Luminex measurements (e.g. Gugatschka et al 2019, Grossmann et al. 2020). For COL1A1, FN1 and TIMP1 measurement, samples were pre-diluted 1:50; to determine all other analytes, samples were undiluted. 

Comment #6

Results: The presentation/ writing will still need to improve. Often times, when reporting certain genes/ proteins have a significant or insignificant change, the authors did not mention the change in relative to which condition/ group. It causes lots of ambiguity in interpreting the results.

Answer #6

The wording of the section results was discussed again and explanations added where we saw any ambiguity (changes are marked in red). The interpretation of the data in combination with the corresponding figures should now be better understandable for the reader.

Comment #7

Discussion. Lines 555-561. The discussion on the roles of IL11 remains brief and superficial. Also, the subsequent sentence of discussing mechanical stimulation and anti-inflammatory effects seemed to be irrelevant. The logical flow of the argument herein needs to be strengthened further.

Answer #7

Thank you for this important comment. We expanded on the roles of IL11. The subsequent sentence was left unmodified since in our opinion it is essential that mechanical stimulation may have anti-inflammatory effects. The following modification was made based on comment #7 in the manuscript, discussion, lines 570-574:

IL11 belongs to the IL6-family. It seems to play a pivotal role in the down-streaming pathway of TGF-�1 and therefore represents an essential component of fibrogenesis, such as in cardiovascular fibrosis. Another recent publication indicates that IL-11 is also a key factor in the development of multiple sclerosis. Consequently, it might be a potential therapeutic target to avoid fibrosis.

Comment #8

Discussion. New studies have revealed the important immunological roles of VF fibroblasts, which should be discussed and cited in the paper. For example:

Foote, Al. G., et al. "Tissue specific human fibroblast differential expression based on RNAsequencing analysis." BMC genomics 20.1 (2019): 308.

King, S. N., et al. "Vocal fold fibroblasts immunoregulate activated macrophage phenotype." Cytokine 61.1 (2013): 228-236.

Li‐Jessen, N. et al. "Cellular source and proinflammatory roles of high‐mobility group box 1 in surgically injured rat vocal folds." The Laryngoscope 127.6 (2017): E193-E200.

Answer #8

Thank you for this comment. We have carefully studied the mentioned publications and included and referenced them in our work. 

The following modifications were made based on comment #8 in the manuscript, introduction and discussion, lines 42-44; lines 112-119 and lines 556-557:

Disruption of the vocal fold (VF) mucosa caused surgically, or mechanically by phonotrauma lead to an upregulation of inflammatory cytokines and enzymes as well as damage associated proteins (DAMP) in the VF.

A special phenomenon of fibroblasts in general is their diversity. They can accomplish a plethora of functions depending on their localization. The studies mentioned above support the idea that fibroblasts respond to vibration by altering their extracellular matrix (ECM) production. The VFF is predominantly responsible for the composition of the lamina propria. Additionally, in a recent study, VFF were shown to be one of the major sources of high-mobility group box 1, an important DAMP. This cell type is, therefore, essential for a smooth phonation as well as for wound healing. 

Particularly, the interaction of VFF and macrophages seem to have a huge impact on the inflammatory response.

Reviewer #2: 

Comment #1 

The new title does not address the concern from the previous review that it was too broad and reads like a review. Please revise this to be study specific. e.g. in vitro mechanical vibration down-regulates pro-inflammatory and pro-fibrotic signalling in human vocal fold fibroblasts.

Answer #1

Thank you for this valuable comment. The title was changed to “In vitro mechanical vibration down-regulates pro-inflammatory and pro-fibrotic signaling in human vocal fold fibroblasts”, based on comment #1.

Comment #2

Nomenclature: In vivo should be italicized on line 393. Human genes and transcripts should be italicized (line 183, 353, 354, 355, 393, 431, 484, 485, 503, 511, 514, 548). Reword line 490 as nomenclature differs between gene and protein - if you list the full name of the gene/protein this would be fine.

Answer #2

All human genes and transcripts were italicized throughout the manuscript based on comment #2 (all changes are marked in red). We could not understand the comment regarding line 490. However, the full names of all genes and proteins are listed at least once in the manuscript. 

Comment #3

Amend line 200 to "After a further 73 hours..."

Answer #3

The following modification was made based on comment #3 in the manuscript, materials and methods, line 213: 

After a further 73 hours, cells were harvested.

Reviewer #3: 

Comment #1

The organization of the Introduction could be improved. Some very short paragraphs were added about the various models; consequently, the revised Introduction does not have good cohesion and flow. I worry that this section seems unfocused: perhaps organization by subheadings would help.

For example,

Injury/Inflammation

Vocal rest (vs. vocalization)

Bioreactor systems

Answer #1

Thank you for this very valuable comment. The paragraphs were organized by the addition of subheadings suggested from reviewer #3 (changes are marked in red).

Comment #2

The statement that therapy “acts primarily as an adjunctive therapy” is not correct with respect to benign lesions [line 55]. Therapy is a primary treatment for nodules and preferred over surgery, for example.

Answer #2

Thank you for this comment. The following modification was made based on comment #2 in the manuscript, introduction, lines 61-66: 

Voice therapy can prevent the development of benign vocal fold lesions and, in some cases (e.g. vocal fold nodules), can also function as primary treatment. However, in the majority of lesions, it acts primarily as an adjunctive therapy to a surgical intervention. In these cases, it is still not clear when it is best to begin with voice therapy, though Tang et al. showed that even a presurgical application has positive effects on subjective voice parameters. 

Comment #3

What is the meaning of the “two most prevalent ECM components” of the VF LP being collagen and HA [line 158]? If “most prevalent” means “most abundant”, then this statement is incorrect. HA is often touted as being functionally important; however, it represents less than 1% of the LP (per total tissue protein). Refer PMID: 16514800.

Answer #3

Thank you for this important comment. We agree with the reviewer, that there is a certain ambiguity in the wording “prevalent”. Therefore, we made the following modification based on comment #2 in the manuscript, materials and methods, lines 173-174:

Two functionally very important ECM components of the VF lamina propria are collagen and HA.

Comments #4 and #5

The Discussion could use attention to organization and flow also. The authors jump back-and-forth between their findings, some clinical studies, animal studies, a bioreactor study – it is challenging to read and grasp the sequence and logic of the authors’ arguments here. It is especially important to take care because the authors shouldn’t overinterpret their data – which reflect in vitro cell phenotypes under certain conditions (admittedly in a sophisticated experimental set-up) – by implying that they are directly comparable to the results of a clinical vocal rest study, for example. Perhaps subheadings would help here also.

The addition of the paragraph on serum starvation is welcome, however it is jarring and seems out-of-place so early in the Discussion. It seems a more natural fit alongside the limitations of the work, later in the Discussion section.

Answers #4 and #5

Thank you for these suggestions. The paragraphs were reordered and the following subheadings were added based on comments #4 and #5 (changes are marked in red): In vivo studies; in vitro studies; Effects of vibration and cytokine-stimuli on VFF.

---

## [Decision Letter · Decision Letter 2]

20 Oct 2020

PONE-D-20-19358R2

In vitro mechanical vibration down-regulates pro-inflammatory and pro-fibrotic signaling in human vocal fold fibroblasts

PLOS ONE

Dear Dr. Hortobagyi,

Thank you for submitting your manuscript to PLOS ONE. After careful consideration, we feel that it has merit but does not fully meet PLOS ONE’s publication criteria as it currently stands. Therefore, we invite you to submit a revised version of the manuscript that addresses the points raised during the review process.

Please address Reviewer 2's comments and resubmit. 

We look forward to receiving your revised manuscript.

Kind regards,

Marie Jetté

Academic Editor

PLOS ONE

Reviewers' comments:

Reviewer's Responses to Questions

**Comments to the Author**

1. If the authors have adequately addressed your comments raised in a previous round of review and you feel that this manuscript is now acceptable for publication, you may indicate that here to bypass the “Comments to the Author” section, enter your conflict of interest statement in the “Confidential to Editor” section, and submit your "Accept" recommendation.

Reviewer #1: All comments have been addressed

Reviewer #2: (No Response)

Reviewer #3: All comments have been addressed

2. Is the manuscript technically sound, and do the data support the conclusions?

Reviewer #1: Yes

Reviewer #2: Yes

Reviewer #3: Yes

3. Has the statistical analysis been performed appropriately and rigorously? 

Reviewer #1: Yes

Reviewer #2: Yes

Reviewer #3: Yes

4. Have the authors made all data underlying the findings in their manuscript fully available?

Reviewer #1: Yes

Reviewer #2: Yes

Reviewer #3: Yes

5. Is the manuscript presented in an intelligible fashion and written in standard English?

Reviewer #1: Yes

Reviewer #2: Yes

Reviewer #3: Yes

6. Review Comments to the Author

Reviewer #1: All comments have been well addressed by the authors. The experimental data are also provided in Supplementary Information.

Reviewer #2: 1. Italicize uses of latin text. The following sections require correction:

Line 479 – italicize in vivo

Line 515 – italicize in vitro

2. The authors edited protein symbols (transcripts = cDNA/RNA, not proteins) during the last revision and they now read as genes. Correct nomenclature for human genes and proteins is as follows:

Gene/RNA/cDNA - italicize and capitalize gene symbols. You are not required to italicize the gene if you write out the full name.

Protein - capitalize protein symbols (no italics)

The Hugo Gene Nomenclature Committee website has a full set of gene names/symbols and the guidelines to follow if further clarification is needed.

The following sections require correction:

Abstract – TGFb, IL1b, aSMA and IL11 are all proteins in this context

Line 41 – 48 – Inflammation section refers to proteins

Line 171 – 187 – ECM related molecules refers to proteins

Line 150 – cytokines are proteins

Lines 179 – 189 – all of the enzymes, fibronectin and HA are proteins

Lines 200/452/525/571/572/574 - aSMA should reflect protein

Line 202 -211 – Should reflect protein

Line 286 – 304 - Western blot detects protein

Lines 313-314 - Luminex detects protein

Line 330 – 332 – ELISA detects protein

Line 366-367 – should reflect protein

Line 377 – HA is protein in Fig 3a.

Line 439 – IL11 and aSMA should reflect protein

Line 505 – italicize IL in IL1b

Line 514 – Branski paper uses cytokine (protein) IL1b

Line 525 – HA and Collagen protein levels measured by ELISA (protein) in Graupp manuscript

Lines 536 – 544 – cytokine/ECM components all proteins

Line 545 – HA should reflect protein

Line 555 – To clarify, in the previous comment addressing line 490 (now 555) I was referring to the convention that by writing a gene name in full, you are not required to italicize it. This allows the term to refer to both gene and protein simultaneously. If you wish to write about changes to gene and protein levels in one sentence, you either have to specify the gene and protein symbols separately, e.g. “FN1 gene and fibronectin protein,” or you could say, “fibronectin gene and protein expression”. My suggestion would be to reword to, “FN1 gene expression and its encoded protein, fibronectin, were upregulated with exposure to cytokines, whereas vibration had no effects.”

Line 559 – COX2 should reflect protein

Line 579-585 – All of these should reflect protein

Line 593 – should reflect proteins

Titles in Fig 3,4,5,and 6 - Amend to reflect gene symbols when mRNA expression was measured (proteins are correct)

Table 1 (Optional) – consider italicizing gene symbols. When a large number of gene symbols are grouped in a table it is optional to italicize per convention.

Reviewer #3: (No Response)

7. PLOS authors have the option to publish the peer review history of their article (what does this mean?). If published, this will include your full peer review and any attached files.

Reviewer #1: No

Reviewer #2: No

Reviewer #3: No

---

## [Author Response · Author response to Decision Letter 2]

21 Oct 2020

Dear Editor and Reviewers,

Thank you again for your effort and time spent to review our manuscript and therefore helping us to improve it. Kindly find below the answer to the comment.

David Hortobagyi

Reviewer #2: 

Comment #1 

The authors edited protein symbols (transcripts = cDNA/RNA, not proteins) during the last revision and they now read as genes. Correct nomenclature for human genes and proteins is as follows:

Gene/RNA/cDNA - italicize and capitalize gene symbols. You are not required to italicize the gene if you write out the full name.

Protein - capitalize protein symbols (no italics)

The Hugo Gene Nomenclature Committee website has a full set of gene names/symbols and the guidelines to follow if further clarification is needed.

Answer #1 

Thank you for this valuable and detailed comment. 

With all genes and proteins the italicized and non-italicized symbols were used, respectively, as suggested from Reviewer#2 and the international nomenclature guidelines (all changes are marked in red).

---

## [Editor Report · Decision Letter 3]

23 Oct 2020

In vitro mechanical vibration down-regulates pro-inflammatory and pro-fibrotic signaling in human vocal fold fibroblasts

PONE-D-20-19358R3

Dear Dr. Hortobagyi,

We’re pleased to inform you that your manuscript has been judged scientifically suitable for publication and will be formally accepted for publication once it meets all outstanding technical requirements.

Kind regards,

Marie Jetté

Academic Editor

PLOS ONE
---

## [Editor Report · Acceptance letter]

28 Oct 2020

PONE-D-20-19358R3 

*In vitro* mechanical vibration down-regulates pro-inflammatory and pro-fibrotic signaling in human vocal fold fibroblasts 

Dear Dr. Hortobagyi:

I'm pleased to inform you that your manuscript has been deemed suitable for publication in PLOS ONE. Congratulations! Your manuscript is now with our production department. 

Kind regards, 

on behalf of

Dr. Marie Jetté 

Academic Editor

PLOS ONE